

SciPost Phys. Lect. Notes 88 (2024)

# The Higgs mechanism with diagrams: A didactic approach

**Jochem Kip and Ronald Kleiss**

Institute for Mathematics, Astrophysics and Particle Physics
Radboud University, Nijmegen, The Netherlands

*Dedicated to the memory of P. W. Higgs* (1929-2024)

## Abstract

We present a pædagogical treatment of the electroweak Higgs mechanism based solely on Feynman diagrams and S-matrix elements, without recourse to (gauge) symmetry arguments. Throughout, the emphasis is on Feynman rules and the Schwinger-Dyson equations; it is pointed out that particular care is needed in the treatment of tadpole diagrams and their symmetry factors.

| | |
|---|---|
| Received | 22-04-2024 |
| Accepted | 30-09-2024 |
| Published | 14-10-2024 |

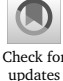

# 1 Introduction

## 1.1 The diagrammatic approach

In particle theory there exist two lines of thought that are well known, but are minority viewpoints. The first is that Feynman diagrams and their Feynman rules are a more fundamental description of the physics than are Lagrangians and actions [1]. The second is that physical requirements like that of unitarity are more fundamental as restrictions on the form of a theory than symmetries that are imposed *a priori* [2,3]. In this light, it becomes interesting to see how the Higgs mechanism (based on spontaneous symmetry breaking (SSB), although we will not explicitly use any symmetry arguments) can be cast into a diagrammatic form without recourse to either Lagrangians or principles of gauge symmetry. This is what we shall explore.

What we shall ultimately derive is the electroweak standard model, so no *new* results are obtained. Rather, it is the *way* in which they are obtained that interests us here: therefore we adopt a pædagogical approach. We shall only consider either scattering amplitudes (S-matrix elements) formed from Feynman diagrams with all external lines on-shell, or *off-shell* amplitudes in which *one* line is kept off-shell.[1] The particular vertices of our models are dictated by the requirement of unitarity, which in the case of massless vector particles means current conservation. The 'fields' of the theory are considered to be labels for bookkeeping of off-shell amplitudes, and particles take on their identity only upon LSZ truncation of external lines [4]. Feynman rules are proposed, not as following from an *a priori* gauge symmetry, but simply in order to make the theory unitary.[2]

---

[1]Connecting two off-shell amplitude results in (a contribution to) an S-matrix element.

[2]That is, we postulate Feynman rules instead of postulating a gauge theory. Inasmuch as physics is the business of proposing rules, and then confront these with experiment, the two approaches are methodologically equivalent.

Since we want to proceed didactically we shall move from simple to more complicated models: therefore the layout of the paper is as follows. We start with a set of self-interacting *tachyons* (the Higgs sector) that end up as a single massive scalar and a number of massless scalars. We then couple the tachyons to a massless vector boson (the Abelian Higgs model) and see how the vector picks up a mass. Subsequently we extend the model to contain three self-interacting vectors (the Apollo model and the Higgs-Kibble model), and then we add a single extra vector to arrive at the electroweak model. Finally, we discuss the inclusion of fermions. A number of technical points are discussed in the appendices.

## 1.2 On unitarity

We have to specify what is meant by 'unitarity' in this paper: we use that term for *partial-wave* unitarity. That is, if in a given $n$-particle scattering amplitude we keep all angles fixed and let the overall energy scale $E$ grow much larger than all masses, the amplitude should asymptotically decrease as $E^{4-n}$ or faster. Loop corrections can modify this behaviour by logarithmic terms at most, but such terms ought to come from exceptional values of the loop momenta, for instance when these become very large, or very collinear with other momenta. Should we restrict also the loop momenta to have magnitude of order $E$ and some fixed direction, then the asymptotic $E^{4-n}$ limit must be respected rigorously.

In a theory with scalar and vector propagators scaling as $E^{-2}$, with minimal coupling between the scalars and the vectors, and with the usual self-interactions between the vectors,[3] unitarity can be proven by simple power counting as long as the external-line factors for the particles go as $E^0$. Problems arise if the vectors are massive: then the appropriate, physical, 'unitary-gauge' propagators[4] for the vectors scale asymptotically as $E^0$, and an external vector's longitudinal polarization vector grows as $E^1$ since at large momentum $p$ it approaches $p/M$, where $M$ is the mass. By showing that such a theory is identical to one in which only massless vectors and tachyons occur, we thus prove that all our models, including the electroweak model in the unitary gauge, satisfy the unitarity requirement.

## 2 Interacting tachyons

### 2.1 Higgs and Goldstones

We define a tachyon to have a *bare* scalar propagator with the 'wrong' mass term. *Free* tachyons may be *physically* unacceptable for reasons of causality, but as far as the Schwinger-Dyson equations (SDe) are concerned their *diagrammatic description* is valid as long as the $i\epsilon$ term has the proper sign [5]. *Self-interacting* tachyons can be physically acceptable as we shall see. We start with $N_t$ ($\geq 2$) tachyons (labelled by $k, \ell, n, \ldots$) with the following Feynman rules:[5]

$$\overrightarrow{\underset{q}{\phantom{xxxx}}}^{n} = \frac{i}{q^2 + m^2/2} \,,$$

$$\underset{p}{\overset{k}{\diagdown}}\!\!\!\!\diagup\!\!\!\!\underset{r}{\overset{n}{\phantom{x}}} = -i\frac{m^2}{v^2}T_{knpr}\,, \quad T_{knpr} = \delta_{kn}\delta_{pr} + \delta_{kp}\delta_{rn} + \delta_{kr}\delta_{np}\,. \tag{1}$$

---

[3]Three-vector vertices going as $E^1$, and four-vector vertices independent of $E$.

[4]The unitary gauge is sometimes, erroneously, described as non-renormalizable.

[5]We shall not explicitly write either $i\epsilon$ or $\hbar$ in what follows.

The term $+m^2/2$ shows the tachyonic character, and $v$ parametrizes the strength of the self-interaction. The tachyons may develop tree-level tadpoles $\tau_n$,[6] described by the SDe[7]

$$\frac{n}{\phantom{x}}\!\!\bullet = \frac{n}{\phantom{x}}\!\!\Big\langle\!\!\bullet\bullet\bullet \,, \tag{2}$$

which we can write out as

$$\tau_n = \frac{2i}{m^2}\left(\frac{-3im^2}{v^2}\frac{\tau_n^3}{3!} + \sum_{\ell \neq n}\frac{-im^2}{v^2}\frac{\tau_n\tau_\ell^2}{2!}\right) = \frac{\tau_n}{v^2}\sum_\ell \tau_\ell^2 \,, \tag{3}$$

so that either all $\tau_n$ vanish (the physically unacceptable solution), or $\tau_n = x_n v$ with $x_n$ the components of a unit vector in $N_t$-dimensional tachyon label space (tl-space):

$$|\tau\rangle = v\,|x\rangle \,, \quad \langle x|x\rangle = 1 \,. \tag{4}$$

The tadpoles will dress the various propagators and mix them, since

$$n\underbrace{\phantom{xxx}}_{}k = \frac{-im^2}{v^2}\left(\delta_{nk}\left(\frac{3\tau_n^2}{2!} + \sum_{\ell \neq n}\frac{\tau_\ell^2}{2!}\right) + (1-\delta_{kn})\,\tau_n\tau_k\right) = -im^2\left(\tfrac{1}{2}\delta_{nk} + x_n x_k\right). \tag{5}$$

The dressed propagators have their own SDe:

$$\Pi = \underbrace{\phantom{xx}}_{}\!\!\oslash\!\!\underbrace{\phantom{xx}}_{} = \underbrace{\phantom{xx}}_{} + \underbrace{\phantom{xx}}_{}\!\!\oslash\!\!\underbrace{\phantom{xx}}_{} \,. \tag{6}$$

Throughout this paper, hatched blobs stand for *connected* diagrams. We give the following steps in detail, since we shall employ them again later on. Multiplying by the denominator $(q^2 + \tfrac{1}{2}m^2)$ we find

$$\Pi_{nk}(q^2 + \tfrac{1}{2}m^2) = i\delta_{nk} + m^2\left(\tfrac{1}{2}\delta_{n\ell} + x_n x_\ell\right)\Pi_{\ell k} \,. \tag{7}$$

Here and in the following, we employ the summation convention: *all* paired labels (in this case, $\ell$) are to be summed over their appropriate range, unless specified otherwise.[8] The terms with $\tfrac{1}{2}m^2$ on either side cancel, so that

$$\Pi_{nk}\,q^2 = i\delta_{nk} + m^2 x_n\,x_\ell\,\Pi_{\ell k} \,. \tag{8}$$

Multiplying both sides by $x_n$ and summing over $n$, we find

$$x_\ell\Pi_{\ell k} = \frac{ix_k}{q^2 - m^2} \,, \tag{9}$$

and we arrive at

$$\Pi_{nk} = \frac{i}{q^2 - m^2}x_n x_k + \frac{i}{q^2}\left(\delta_{nk} - x_n x_k\right). \tag{10}$$

We can choose a complete orthonormal basis in tl-space:

$$\langle x|x\rangle = 1 \,, \quad \langle y^j|y^k\rangle = \delta_{jk} \,, \quad \langle x|y^j\rangle = 0 \,, j,k \in \{2,\dots,N_t\} \,, \tag{11}$$

so that

$$\delta_{nk} - x_n x_k = (y^j)_n(y^j)_k \,. \tag{12}$$

---

[6]Since Lorentz invariance forbids tadpoles for non-zero spin, SSB requires the presence of scalar particles.

[7]In Lagrangian-speak, this is the Euler-Lagrange equation for vanishing momentum.

[8]It is under-appreciated that the summation convention is naturally connected with the diagrammatic approach, since a line has precisely two endpoints.

We now introduce the concept of *active* and *passive* vertices. The active vertices are those in which at least two of the momenta involved are linearly independent; the vertex of Eq.(5) is passive, not active. In every diagram that contributes to a scattering amplitude any internal line must end in active vertices somewhere;[9] therefore, concentrating on a particular internal line, we can write an amplitude as

$$\mathcal{M} = A_n \Pi_{nk} B_k\,. \tag{13}$$

$A_n$ is the active off-shell amplitude emitting $n$, and $B_k$ is the one absorbing $k$.[10] The active vertices are indicated by dots. We see that we can write

$$\mathcal{M} = A_h R_h(q) B_h + A_j R_0(q) B_j\,, \tag{14}$$

where

$$R_h(q) = \frac{i}{q^2 - m^2}\,, \qquad R_0(q) = \frac{i}{q^2}\,, \qquad A_h = A_n x_n\,, \qquad A_j = A_n (y^j)_n\,, \tag{15}$$

and similarly for $B$: the internal line therefore represents one massive propagator and a bunch of $N_t - 1$ massless ones, the so-called Goldstone bosons. By letting $A$ and $B$ move very far apart in spacetime, so that the internal line is truncated [5], we identify $A_h$ as the source emitting a Higgs scalar of mass $m$. We can also determine the active vertices of the reformulated theory. For instance,

$$-i\frac{m^2}{v^2}\, T_{knpr}\, x_k x_n x_p x_r = -3i\frac{m^2}{v^2}\,,$$

$$= -3i\frac{m^2}{v}\,, \tag{16}$$

and in a similar way we find

$$= -i\frac{m^2}{v^2}\left(1 + 2\delta_{jk}\right)\,, \qquad = -i\frac{m^2}{v^2}\,, \qquad = -i\frac{m^2}{v}\,. \tag{17}$$

A few observations are important at this point. In the first place, the identity (2) is precisely that, an *identity*. Therefore we must use

either ⬤—● or ⬤⦃● but *not* both.

This counter-intuitive-seeming prescription is the basis for the diagrammatic description of the Higgs mechanism. Not taking Eq.(2) as an actual physical identity leads to incorrect handling of the symmetry factors, and possible mis-counting of diagrams. For example, iterating one leg of the SDe (2) would give

$$\tag{18}$$

with the incorrect implication that $\sum_n \tau_n^2 = v^2/3$. Only if we iterate all legs,

$$\tag{19}$$

do we again find the correct result.

---

[9] This excludes vacuum diagrams; see below.

[10] The blobs $A$ and $B$ may be connected by other lines, in which case we have a loop diagram.

Secondly, if we treat the tadpoles diagrammatically we have to admit that they are zero modes, *i.e.* they are constant. If we let $\tau_n$ depend on position, the tadpoles act as sources of momentum that will needlessly tangle our treatment. We therefore take $|x\rangle$ to be a *constant* unit vector.

Finally, the very *idea* of SSB is that $|x\rangle$ is also a *random* unit vector, outside of our control: we are therefore forbidden from making any further assumptions on $|x\rangle$. Of course, a good model ought to yield physics that is, as much as possible, independent of $|x\rangle$.

## 2.2 A Goldstone infrared problem

In the foregoing we have derived a consistent set of Feynman rules containing one massive scalar and at least one massless Goldstone scalar. Once we try to truncate the massless internal lines, however, we run into problems for $N_t > 2$, since it is not clear how we can disentangle the various massless propagators so as to identify the various Goldstones. Worse, the model contains infrared phenomena. Let us imagine a process in which a massless Goldstone is emitted with momentum $p$:

$$\mathcal{M}_0(p) = \;\; \rlap{\text{(diagram)}} \;\; p \,. \tag{20}$$

The (differential) cross section is given by

$$d\sigma_0 \sim |\mathcal{M}_0(p)|^2 \, d^4p \, \delta(p^2)\theta(p^0)\,. \tag{21}$$

We may let the Goldstone go very slightly off-shell and decay into three. Assuming that $\mathcal{M}_0(p)$ does not depend too drastically on $p$, we can write this as

$$\mathcal{M}_1 = \;\; \rlap{\text{(diagram)}} \;\; \approx \mathcal{M}_0(q_1 + q_2 + q_3)\frac{1}{(q_1 + q_2 + q_3)^2}\frac{m^2 k}{v^2}\,, \tag{22}$$

where $k = 3$ if the three Goldstones are identical, otherwise 1. The cross section, which is dominated by the diagram of Eq.(22), can then be written as

$$d\sigma_1 \sim |\mathcal{M}_1|^2 \prod_{j=1}^{3} d^4q_j \, \delta(q_j^2)\,\theta(q_j^0)\delta^4(q_1 + q_2 + q_3 - p)\, d^4p \, \delta(p^2 - u)\, du$$

$$\approx |\mathcal{M}_0(p)|^2 d^4p \, \delta(p^2 - u)\,\theta(p^0)\frac{m^4 k}{2v^4}\frac{\pi^2}{8}\frac{du}{u} \approx d\sigma_0 \frac{m^4 k}{2v^4}\frac{\pi^2}{8}\frac{du}{u}\,, \tag{23}$$

where we have used the result [5]

$$\int \left(\prod_{j=1}^{3} d^4q_j \, \delta(q_j^2)\,\theta(q_j^0)\right)\delta^4(q_1 + q_2 + q_3 - p) = \frac{\pi^2}{8}p^2\,. \tag{24}$$

We see that the $u$ integral in Eq.(23) diverges for $u \to 0$: an infrared divergence.[11] Physically, this means that an external massless Goldstone in any process will unavoidably emerge as a cloud of low-energy Goldstones, all moving collinearly with the speed of light. This makes the LSZ truncation for such particles extremely problematic. It would seem that if we want to end up with a model in which particles can be unambiguously identified, massless Goldstone bosons are to be avoided.[12]

---

[11] In a good theory such as QED, the actual IR divergences are cancelled by corresponding IR divergences from virtual corrections. Even if this happens here, the fact remains that for very small $u$ a three-body-decaying Goldstone wins out over a non-decaying one.

[12] One might also worry about internal lines carrying vanishing momentum, for instance in the vacuum diagram ◯—◯. However, some reflection shows that such zero-momentum lines can only be the massive Higgs propagators.

## 3  The Abelian Higgs model

### 3.1  Feynman rules and current conservation

We may couple the tachyons to a massless vector (called 'photon' for now), with a propagator given by

$$\alpha \,\rasll\, \beta = -\frac{i}{q^2} k^{\alpha\beta}\,, \quad k^{\alpha\beta} = g^{\alpha\beta} + F^{\alpha}q^{\beta} + q^{\alpha}F^{\beta} + G\,q^{\alpha}q^{\beta}\,. \tag{25}$$

The quantities $F^{\mu}$ and $G$ are related to the gauge choice: in this model, they are quite immaterial as we shall see. We shall assume two interaction vertices between photon and tachyons:

$$\mu \,\rasll\, \begin{matrix}p \nearrow n\\ q \searrow k\end{matrix} = f^{n}_{k}(p-q)^{\mu}\,, \quad \begin{matrix}\alpha\\ \beta\end{matrix}\,\rasll\,\begin{matrix}n\\ k\end{matrix} = i\,t^{n}_{k}\,g^{\alpha\beta}\,. \tag{26}$$

Because the tachyons are bosons, the real matrices $f$ and $t$ must of necessity be *antisymmetric* and *symmetric*, respectively.[13]

Since the photon is massless, its *observable* polarization must be purely transverse in *any* Lorentz frame. That is only possible if its current is conserved [5]. As before, we shall give the following steps in some detail since we will employ them again later on. Let $M(q)^{\mu}$ be the complete set of connected diagrams that emit or absorb a photon with momentum $q$ (not necessarily on-shell):

$$M(q)^{\mu} = \bigcirc\!\!\!\!\!\!\!\!\rasll^{\,q}\, \mu\,. \tag{27}$$

The requirement of current conservation means that

$$M(q)^{\mu}\,q_{\mu} = \bigcirc\!\!\!\!\!\!\!\!\rasll\bullet \overset{!}{=} 0\,. \tag{28}$$

The symbol $\overset{!}{=}$ means that we *demand* the zero result: we have to construct our theory so as to arrange it. The 'handlebar' denotes multiplication with the momentum, considered *outgoing*. We have to prove Eq.(28) in full generality, and we shall do this using the SDe. Let us denote by a lightly shaded *semi-connected* blob a complete set of diagrams that are not necessarily connected but do not contain vacuum bubbles. Now consider

$$\mu \,\rasll\, \begin{matrix}p \nearrow\\ n\\ k\\ q \searrow\end{matrix}\!\!\!\bigg) = \frac{i}{p^2 + m^2/2}\, f^{n}_{k}(p-q)^{\mu}\,\frac{i}{q^2 + m^2/2}\,, \tag{29}$$

where we have left out the expression for the semi-connected blob. Applying the handlebar gives us

$$\begin{aligned}\rasll\,\begin{matrix}p \nearrow\\ n\\ k\\ q \searrow\end{matrix}\!\!\!\bigg) &= \frac{i}{p^2 + m^2/2}\, f^{n}_{k}(p-q\cdot-p-q)\,\frac{i}{q^2 + m^2/2}\\ &= \frac{i}{p^2 + m^2/2}\, f^{n}_{k}\,i + i\,f^{k}_{n}\,\frac{i}{q^2 + m^2/2}\,. \end{aligned} \tag{30}$$

---

[13] If current conservation (see further on) is to have any chance at all, the combination $(p-q)^{\mu}$ is unavoidable, as is therefore the antisymmetry of $f$.

Note that this works as long as the mass terms (the $+m^2/2$) are the same in both tachyon propagators: their sign is irrelevant. We can represent Eq.(30) diagrammatically by auxiliary Feynman rules, as follows:

$$ \text{(diagram)} = \text{(diagram)}, \quad \text{(diagram)}^n_k = f^n_k, \quad \text{(diagram)} = i. \tag{31} $$

It is important to note that, unless we specify them explicitly, equivalent lines entering the semiconnected blob have to be symmetrized over, and therefore Eq.(31) includes *both* terms of Eq.(30). In addition, if a slashed line happens to be an external one, it will not survive LSZ truncation since it has no pole: therefore such diagrams can be neglected.[14] Next, we consider the following tree-level amplitude:[15]

$$ \text{(diagram)} = \text{(diagram)} + \text{(diagram)} + \text{(diagram)} \overset{!}{=} 0. \tag{32} $$

Applying the Feynman rules we find

$$ -i\left(2(f^2)^n_k + t^n_k\right)(p+q+r)^\alpha \overset{!}{=} 0, \tag{33} $$

so that we require

$$ t^n_k \overset{!}{=} -2 f^n_\ell f^\ell_k = 2 f^n_\ell k^k_\ell. \tag{34} $$

We then have

$$ \text{(diagram)} = - \text{(diagram)}. \tag{35} $$

The proof of general current conservation proceeds as follows, where we leave the semi-connected blobs on the right to be understood. The SDe reads

$$ \text{(diagram)} = \text{(diagram)} + \text{(diagram)}. \tag{36} $$

We now apply the handlebar. Since slashed external lines do not contribute, we can SDe-iterate the slashed propagators by letting them enter a vertex, so that

$$ \text{(diagram)} = - \text{(diagram)}, $$
$$ \text{(diagram)} = + \text{(diagram)} + \text{(diagram)} + \text{(diagram)}. \tag{37} $$

Let $n$, $k$ and $r$ be distinct tachyon labels. Then, dropping unimportant overall factors (denoted by $\sim$), we have

$$ \text{(diagram)} = \text{(diagram)} + \text{(diagram)} \sim (f^3)^n_k + (f^3)^k_n = 0, \tag{38} $$

$$ \text{(diagram)} = \text{(diagram)} + \text{(diagram)} + \text{(diagram)} + \text{(diagram)} $$
$$ \sim \left(\frac{3}{3!} f^k_n + \frac{1}{2!} f^n_k\right) + \sum_{r \neq k,n} \left(\frac{1}{2!} f^n_k + \frac{1}{2!} f^k_n\right) = 0. \tag{39} $$

---

[14]This is because we consider S-matrix elements rather than Green's functions.

[15]Written like this, the process is kinematically impossible. By moving tachyon $n$, say, to the initial state (replacing $p$ by $-p$) we can repair this. The conclusion remains the same.

This finishes the proof of current conservation, Eq.(28): the only condition on $f$ is that it be antisymmetric. The presence of tadpoles does not change the argument, since the semi-connected blob may contain tadpoles at will. We simply must adhere to the convention of

$$\text{never using } \bigcirc\!\!\!/\!\!-\!\!\bullet \text{ but always } \bigcirc\!\!\!/\!\!\!\!<\!\!\bullet^{\bullet} \, . \tag{40}$$

That is, in the SDe iteration the tadpole does not count as a vertex.

## 3.2 Dressed propagators and masquerading

Next, we turn to the dressed propagators, the analogues of $\Pi_{nk}$ of the previous section. We encounter two additional passive vertices, in addition to that of Eq.(5): introducing another vector in tl-space,

$$e_n = x_\ell f_n^\ell, \quad |e\rangle = f\,|x\rangle, \quad \langle e|e\rangle \equiv e^2, \quad \langle e|x\rangle = 0, \tag{41}$$

we have

$$\begin{aligned} &n\underset{\alpha}{\overset{q}{\rightsquigarrow}} = \tau_\ell f_n^\ell q^\alpha = v\,e_n\,q^\alpha, \\ &\alpha\rightsquigarrow\beta = \frac{1}{2!}\tau_k\tau_n t_k^n g^{\alpha\beta} = iv^2\langle e|e\rangle g^{\alpha\beta} = i\,e^2 v^2 g^{\alpha\beta}\,. \end{aligned} \tag{42}$$

There are now three types of dressed propagators, with the momentum $q$ assumed to be moving from left to right:

$$\begin{aligned} \Pi_{nk} &= n\!-\!\!\bigcirc\!\!-\!k = -\!\!-\!- + \overset{\bullet\bullet}{\rightsquigarrow}\!\!\bigcirc\!\!- + -\!\!\rightsquigarrow\!\!\bigcirc\!\!-\,, \\ \Psi_k &= \rightsquigarrow\!\!\bigcirc\!\!-\!k = \rightsquigarrow\!\!\bigcirc\!\overset{\bullet\bullet}{\rightsquigarrow} + \rightsquigarrow\!\!\bigcirc\!\!\rightsquigarrow\,, \\ \Omega &= \rightsquigarrow\!\!\bigcirc\!\!\rightsquigarrow = \rightsquigarrow\!\!\!- + \rightsquigarrow\!\!\bigcirc\!\overset{\bullet\bullet}{\rightsquigarrow}\!\!\rightsquigarrow + \rightsquigarrow\!\!\bigcirc\!\!\rightsquigarrow\,.^{16} \end{aligned} \tag{43}$$

Following the steps described in section 2.1 we can write[17]

$$\begin{aligned} \Pi_{nk}\,q^2 &= i\delta_{nk} + m^2 x_n x_\ell\,\Pi_{\ell k} + i\nu\,e_n\,q^\mu\Psi_k^\mu\,, \\ \Psi_k^\alpha\,q^2 &= m^2\,\Psi_\ell^\alpha x_\ell x_k - i\nu\,\Omega^{\alpha\mu}q^\mu\,e_k\,, \\ \Omega^{\alpha\beta}\,q^2 &= -ik^{\alpha\beta} + e^2 v^2\,\Omega^{\alpha\mu}k^{\mu\beta} - i\nu\,\Psi_\ell^\alpha\,e_\ell\,q^\mu k^{\mu\beta}\,, \end{aligned} \tag{44}$$

so that

$$x_\ell\,\Pi_{\ell k} = \frac{i\,x_k}{q^2 - m^2}\,, \qquad \Psi_\ell^\alpha x_\ell = 0\,. \tag{45}$$

Introducing an orthonormal basis in tl-space (and ignoring the possible interpretational problem raised in section 2.2):

$$|x\rangle\,, \frac{1}{e}\,|e\rangle\,, \left|y^j\right\rangle \quad (j=3,\dots,N_t) \;\Rightarrow\; \delta_{nk} - x_n x_k = \frac{1}{e^2}e_n e_k + (y^j)_n(y^j)_k\,, \tag{46}$$

we arrive at the forms

$$\begin{aligned} \Pi_{nk} &= \frac{i}{q^2 - m^2}x_n x_k + \frac{i}{q^2}\left(\frac{1}{e^2}e_n e_k + y_n^j y_k^j\right) + \frac{v^2}{q^4}e_n q^\lambda\,\Omega^{\lambda\sigma}q^\sigma\,e_k\,, \\ \Psi_k^\alpha &= -\frac{iv}{q^2}\,\Omega^{\alpha\lambda}q^\lambda\,e_k\,, \end{aligned} \tag{47}$$

$$\Omega^{\alpha\beta}\,q^2 = -ik^{\alpha\beta} + M^2\,\Omega^{\alpha\mu}T^{\mu\nu}k^{\nu\beta}\,, \tag{48}$$

---

[16]$\Pi$ and $\Omega$ are even in $q$, but $\Psi$ is *odd* in $q$.

[17]For typographical reasons we shall always write Lorentz indices as upper indices, and the Minkowski metric is understood.

with

$$M = ev, \qquad T^{\alpha\beta} = g^{\alpha\beta} - L^{\alpha\beta}, \qquad L^{\alpha\beta} = q^\alpha q^\beta / q^2. \tag{49}$$

We cannot solve Eq.(48) for $\Omega$, but as it will turn out that does not matter. Another point worthy of note follows from the expression for $\Pi_{nk}$. As we can see from its diagrammatic SDe, the result for $x_\ell \Pi_{\ell k}$ is the same as in Eq.(9). This is because the scalar-vector mixing tadpole contains a vector $|e\rangle = f|x\rangle$, which is orthogonal to $|x\rangle$. Introducing several vector particles will lead to different vectors $|e\rangle$, but these are also orthogonal to $|x\rangle$ (see later on). Thus, the propagator $\Pi_{nk}$ will always contain the term $ix_n x_k/(q^2 - m^2)$, in other words *there will always be a distinct Higgs particle*; this is the original assertion made in [6].

A final ingredient is the following. Again using active vertices, we have two SDe's:

$$\text{(diagram)} \tag{50}$$

Current conservation can now be expressed by the handlebar diagrams:

$$0 = \text{(diagram)} \tag{51}$$

$$\text{(diagram)} \tag{52}$$

Everything except the active-vertex diagrams cancels.[18] Denoting by $A_\gamma^\mu$ the amplitude emitting the photon and by $A_n$ the one emitting tachyon $n$, we see that current conservation implies

$$\text{(diagram)} = 0 \quad \Rightarrow \quad A_n e_n = \frac{i}{v} A_\gamma^\mu q^\mu, \tag{53}$$

again with the momentum $q$ counted outgoing. In this way a combination of tachyonic tadpoles can *masquerade* as a handlebarred vector boson.

## 3.3 Amplitudes

We are now ready to consider an amplitude, as in the previous section:

$$\mathcal{M} = \text{(diagram)}$$
$$= A_n \Pi_{nk} B_k + A_\gamma^\alpha \Psi_k^\alpha B_k + A_n(-\Psi_n^\beta)B_\gamma^\beta + A_\gamma^\alpha \Omega^{\alpha\beta} B_\gamma^\beta. \tag{54}$$

Using the results (47) and Eq.(53) we can write this as

$$\mathcal{M} = A_h R_h(q) B_h + A_j R_0(q) B_j + A_\gamma^\alpha R_M^{\alpha\beta}(q) B_\gamma^\beta, \tag{55}$$

where

$$R_M^{\alpha\beta}(q) = \frac{i}{M^2} L^{\alpha\beta} + D^{\alpha\beta}, \quad D^{\alpha\beta} = T^{\alpha\mu} \Omega^{\mu\nu} T^{\nu\beta}. \tag{56}$$

Since $T^{\alpha\mu} k^{\mu\nu} T^{\nu\beta} = T^{\alpha\beta}$, Eq.(48) implies

$$D^{\alpha\beta} q^2 = -i T^{\alpha\beta} + M^2 D^{\alpha\beta}, \tag{57}$$

---

[18]To appreciate this, it may help to recast diagrams:

$$\text{(diagram)} = 0,$$

where we can recognize simply a special case of Eq.(39). Notice the importance of rule (40)!

and so we find for the vector propagator

$$R_M^{\alpha\beta}(q) = \frac{i}{M^2} L^{\alpha\beta} - \frac{i}{q^2 - M^2} T^{\alpha\beta} = \frac{i}{q^2 - M^2}\left(-g^{\alpha\beta} + \frac{q^\alpha q^\beta}{M^2}\right),\tag{58}$$

the *unitary-gauge* propagator. As announced, the quantities $F^\mu$ and $G$ of Eq.(25) drop out because $q^\alpha T^{\alpha\beta} = 0$. Upon LSZ truncation, the massive vector has precisely its three physical polarization degrees of freedom, including the longitudinal one.

## 3.4  Vertices

The vertices of Eq.(17) are unchanged, but we have to find the scalar-vector interactions. We see immediately that

$$\sim \langle x|f|x\rangle = 0, \qquad \sim \langle x|f|y^j\rangle \sim \langle e|y^j\rangle = 0.\tag{59}$$

The two-photon vertices[19] require more care. First, we find

$$= -2i\langle x|t|x\rangle\, g^{\alpha\beta} = 2i\,e^2\,g^{\alpha\beta} = 2i\frac{M^2}{v^2}g^{\alpha\beta}, \qquad = 2i\frac{M^2}{v}g^{\alpha\beta}.\tag{60}$$

However,

$$\sim \langle x|t|y^j\rangle \sim \langle e|f|y^j\rangle.\tag{61}$$

If we want this to vanish we need $f|e\rangle \sim |x\rangle$. As can be seen from appendix A, that is only possible if $|x\rangle$ is an eigenvector of $f^2$: but $|x\rangle$ is random! The only reasonable solution is to require that $|x\rangle$ is *always* an eigenvector, *i.e.* $f^2$ must be $-e^2$ times the unit matrix, and the number of tachyons must be even. In that case, the vectors $|y^j\rangle$ can be chosen such that

$$f\left|y^{2s-1}\right\rangle = e\left|y^{2s}\right\rangle, \quad f\left|y^{2s}\right\rangle = -e\left|y^{2s-1}\right\rangle, \quad s = 2,\dots,N_t/2,\tag{62}$$

so that they can be interpreted as the real and imaginary part of charged scalars, with

$$= e\,(p-q)^\mu, \qquad = 2ie^2\,g^{\alpha\beta}.\tag{63}$$

If $f^2$ is not proportional to unity, we have a model that is consistent and unitary, but lacks an interpretation in terms of simple charged scalars. Admittedly, since the physics in that case depends on $|x\rangle$, it is not a 'good' model in the sense discussed above.

For $N_t = 2$, the 'original' AH model, the Feynman rules found are precisely those of the sector of the Standard Electroweak Model that contains only $Z^0$ and Higgs particles. Since the longitudinal polarization vector of a $Z^0$ with momentum $p$ has the form $\epsilon_L^\mu = p^\mu/M - \mathcal{O}(M/p^0)$, the application of Eq.(53) shows that the longitudinal degree of freedom is essentially scalar tachyons in masquerade,[20] and so unitarity of this sector is proven.

---

[19]For reasons lost in the mists of time, these are also called *seagull vertices*.

[20]The fact that longitudinally polarized vector bosons may be replaced by appropriate (combinations of) scalars is referred to as the *Equivalence Theorem* [7].

# 4 Interacting vectors: General structure

## 4.1 Feynman rules and current conservation

We now turn to models containing several massless vectors (labelled by $w, x, y, \ldots$), that may be interacting with each other. It becomes necessary to postulate their propagator in more detail:[21]

$$\alpha \overset{q}{\underset{w}{\leadsto}} \beta = R_n^{\alpha\beta}(q) = -i\frac{k^{\alpha\beta}(q)}{q^2}, \quad k^{\alpha\beta}(q) = g^{\alpha\beta} - \frac{q^\alpha n^\beta}{(q \cdot n)} - \frac{n^\alpha q^\beta}{(q \cdot n)} + \frac{n^2 q^\alpha q^\beta}{(q \cdot n)^2}, \tag{64}$$

where $n^\alpha$ is a fixed vector. This is the *axial gauge*, with $n^\alpha k^{\alpha\beta} = 0$. We assume a three-vector coupling, defined by

$$\lambda \underset{q}{\overset{p}{\leadsto}} \overset{\alpha}{\underset{\beta}{\overset{x}{\underset{y}{\leadsto}}}} = h^{wxy} Y(p, \alpha; q, \beta; -p-q, \lambda), \,^{22} \tag{65}$$

with

$$Y(p, \alpha; q, \beta; r, \lambda) = (p-q)^\lambda g^{\alpha\beta} + (q-r)^\alpha g^{\beta\lambda} + (r-p)^\beta g^{\lambda\alpha}. \tag{66}$$

Since the vectors are bosons, the real numbers $h^{wxy}$ are necessarily totally antisymmetric. We also propose a four-vector interaction:

$$\underset{v}{\overset{\alpha}{\underset{w}{\leadsto}}} \overset{x}{\underset{z}{\overset{\beta}{\underset{\mu}{\leadsto}}}} = i X(w, \alpha; x, \beta; y, \mu; z, \nu), \tag{67}$$

with

$$X(w, \alpha; x, \beta; y, \mu; z, \nu) = g^{\alpha\beta} g^{\mu\nu}\{wyzx\} + g^{\alpha\mu} g^{\beta\nu}\{wxzy\} + g^{\alpha\nu} g^{\beta\mu}\{wxyz\},$$
$$\{wxyz\} = h^{wxt} h^{yzt} + h^{wyt} h^{xzt}. \tag{68}$$

We need to work out what the handlebar does here:

$$\overset{p}{\underset{q}{\overset{w}{\leadsto}}} \overset{x}{\underset{y}{\leadsto}} \sim R_n^{\mu\alpha}(p) h^{wxy} Y(p, \alpha; q, \beta; -p-q, \lambda)(-p-q)^\lambda R_n^{\beta\nu}(q). \tag{69}$$

Using the two results

$$Y(p, \alpha; q, \beta; -p-q, \lambda)(-p-q)^\lambda = \left(p^\alpha p^\beta - p^2 g^{\alpha\beta}\right) - \left(q^\alpha q^\beta - q^2 g^{\alpha\beta}\right),$$
$$R_n^{\mu\alpha}(p)\left(p^\alpha p^\beta - p^2 g^{\alpha\beta}\right) = i\left(g^{\mu\alpha} - \frac{p^\mu n^\alpha}{(p \cdot n)}\right) g^{\alpha\beta}, \tag{70}$$

we find immediately the rule

$$\overset{}{\underset{}{\leadsto}} = \cdots \overset{}{\underset{}{\leadsto}}, \quad \overset{w}{\underset{y}{\leadsto}} \overset{\alpha}{\underset{\beta}{\overset{x}{}}} = h^{wxy} g^{\alpha\beta}, \quad \alpha \leadsto \beta = i g^{\alpha\beta}. \,^{23} \tag{71}$$

---

[21]We do not have to 'derive' this propagator by 'gauge fixing' in some action, since we do not use actions. The only requirement is that our resulting theory be current-conserving: recall footnote 2.

[22]This is just the tachyon-tachyon-vector vertex, adapted to three vectors, that have to be treated on the same footing (being bosons). It is therefore essentially the simplest possible choice.

[23]There is a subtlety if one of the vectors is external. Appendix B shows that this does not lead to problems.

In addition, we have

$$\text{(diagram)} = - \text{(diagram)}, \tag{72}$$

which is the *raison d'être* of the cumbersome expression (68). For the interactions with the tachyons, we simply generalize those of the AH model:

$$\mu \overset{p}{\underset{q}{\overset{w}{\gtrless}}}{}^n_k = (f^w)^n_k (p-q)^\mu, \qquad \alpha \overset{w}{\underset{\beta}{\overset{}{\gtrless}}}{}^n_{x}{}_k = i \, (t^{wx})^n_k \, g^{\alpha\beta}, \tag{73}$$

with again $f^w$ antisymmetric and $t^{wx} = t^{xw}$ symmetric in tl-space. By examing the tree-level handlebar condition

$$\overset{w}{\underset{n}{\overset{}{\gtrless}}}{}^k_x = \overset{w}{\gtrless}{}^k_x{}_n + \overset{w}{\gtrless}{}^k_x{}_n + \overset{w}{\gtrless}{}^x_k{}_n + \overset{w}{\gtrless}{}^k_x{}_n \overset{!}{=} 0, \tag{74}$$

we find the analogue to Eq.(28):

$$[f^w, f^x] \overset{!}{=} h^{wxy} f^y, \qquad \{f^w, f^x\} \overset{!}{=} -t^{wx}. \tag{75}$$

With these rules and tools we can prove current conservation for this type of model. Again leaving out the semi-connected blobs, we write the SDe as

$$\text{(diagram)} = \text{(diagram)} + \text{(diagram)} + \text{(diagram)} + \text{(diagram)}, \tag{76}$$

and we have

$$\text{(diagram)} = - \text{(diagram)} - \text{(diagram)},$$

$$\text{(diagram)} = + \text{(diagram)} + \text{(diagram)} + \text{(diagram)},$$

$$\text{(diagram)} = - \text{(diagram)},$$

$$\text{(diagram)} = + \text{(diagram)} + \text{(diagram)} + \text{(diagram)} + \text{(diagram)}. \tag{77}$$

By methods similar to those leading to Eq.(39) we can straightforwardly show that

$$\text{(diagram)} = 0, \qquad \text{(diagram)} + \text{(diagram)} = 0, \tag{78}$$

admittedly with extensive use of Eq.(75). We see that the currents for the massless vectors are strictly conserved.

## 4.2 Dressed propagators, masquerading and the amplitude

The passive vertices of the model are generalizations of those of the AH model: in addition to Eq.(5) we have

$$
\begin{aligned}
n\xrightarrow{\;\;q\;\;}\alpha &= \tau_\ell\,(f^w)_n^\ell\,q^\alpha \;=\; \nu\,(e^w)_n\,q^\alpha\,, \\
\alpha \cdots \beta &= \frac{1}{2!}\,\tau_n\,\tau_k\,(t^{wx})_k^n\,g^{\alpha\beta} \;=\; i\,\nu^2\,\langle e^w|e^x\rangle\,g^{\alpha\beta}\,,
\end{aligned}
\tag{79}
$$

with $|e^w\rangle = f^w|x\rangle$, or more explicitly $(e^w)_n = (f^w)_n^k x_k$. The SDe's for the dressed propagators:

$$
\begin{aligned}
\Pi_{nk} &= n\;\multimap\!\!\circledcirc\!\!\multimap\; k = \rule{1em}{0.4pt} + \;\Yup\!\!\circledcirc\!\!\rule{0.8em}{0.4pt}\; + \;\rule{0.5em}{0.4pt}\!\!\circledcirc\!\!\Yup\;, \\
\Psi_{wk} &= \;\sim\!\!\circledcirc\!\!\multimap\; k = \;\sim\!\!\circledcirc\!\!\Yup\; + \;\sim\!\!\circledcirc\!\!\rule{0.5em}{0.4pt}\;, \\
\Omega_{wx} &= \;\sim\!\!\circledcirc\!\!\sim\; = \;\sim\!\!\sim\; + \;\sim\!\!\circledcirc\!\!\Yup\!\!\sim\; + \;\sim\!\!\circledcirc\!\!\rule{0.5em}{0.4pt}\;,
\end{aligned}
\tag{80}
$$

can be treated in the same way as before, to yield

$$
\begin{aligned}
\Pi_{nk} &= \frac{i}{q^2-m^2}x_n x_k + \frac{i}{q^2}(\delta_{nk}-x_n x_k) + \frac{\nu^2}{q^4}(e^w)_n q^\mu \Omega_{wx}^{\mu\nu} q^\nu (e^x)_k\,, \\
\Psi_{wk}^\alpha &= -\frac{i\nu}{q^2}\Omega_{wy}^{\alpha\mu} q^\mu (e^y)_k\,, \\
\Omega_{wx}^{\alpha\beta} q^2 &= -ik^{\alpha\beta}\delta_{wx} + \nu^2\,\Omega_{wy}^{\alpha\mu} T^{\mu\nu} k^{\nu\beta}\,\langle e^y|e^x\rangle\,.
\end{aligned}
\tag{81}
$$

If we denote by $A_w^\mu$ the active-vertex source emitting vector $w$, we can perform the same steps as for Eq.(53) and find the masquerading identities

$$
A_n(e^w)_n = \frac{i}{\nu}A_w^\mu q^\mu\,,
\tag{82}
$$

that we shall use extensively. We do have to note the appearance of an extra term on the right-hand side in Eq.(51):

$$
-\;\circledcirc\!\!\sim\!\!\prec\cdots
$$

but since the tadpoles carry no momentum this diagram vanishes by itself.

The amplitude, with dressed propagators running between active vertices, is also treated as before, and we find

$$
\mathcal{M} = A_h R_h(q) B_h + A_n \frac{1}{q^2}(\delta_{nk}-x_n x_k) B_k + A_w^\alpha D_{wx}^{\alpha\beta} B_x^\beta\,,
\tag{83}
$$

with

$$
D_{wx}^{\alpha\beta} = T^{\alpha\mu}\Omega_{wx}^{\mu\nu} T^{\nu\beta} = -iT^{\alpha\beta} K_{wx}\,, \quad K_{wx}q^2 = \delta_{wx} + \nu^2 K_{wy}\,\langle e^y|e^x\rangle\,.
\tag{84}
$$

Further evaluation depends on the details of the model: in the following we shall examine several such models. It may be interesting to point out that the choice of the axial propagator of Eq.(64) is necessary to make Eq.(71) work, but that is the *only* place in the whole discussion where it plays a rôle: in particular the propagators $D^{\alpha\beta}$ are independent of the gauge choice.[24]

---

[24]And the axial-gauge vector $n^\mu$ disappears completely from our discussion, as it *ought to*.

# 5 Interacting vectors: Example models

## 5.1 The Apollo model

The minimum number $N_v$ of vector fields that can be self-interacting is 3, in which case $h^{wxy}$ must be proportional to the Levi-Civita symbol $\varepsilon^{wxy}$. Since the $f$ matrices are antisymmetric, the minimum number $N_t$ of tachyons is also 3.[25] Therefore the Apollo model (first discussed in [8]), with $N_t = N_v = 3$, is the 'smallest' model with vector self-interactions. Let us define the $3 \times 3$ matrices $f^w$ for this model by

$$(f^w)^n_k = -e \, O^w_y \, \varepsilon^{ynk} \,, \tag{85}$$

where $O$ is an arbitrary but fixed orthogonal $3 \times 3$ matrix. It can be checked that, indeed,

$$[f^w, f^x] = h^{wxy} f^y \,, \quad h^{wxy} = e \, \varepsilon^{wxy} \,. \tag{86}$$

Since $\langle e^w | x \rangle = 0$, the three vectors $|e^w\rangle$,

$$(e^w)_k = e \, O^w_y \varepsilon^{ykn} x_n \,, \quad w = 1, 2, 3 \,, \tag{87}$$

cannot be linearly independent. If we define

$$\gamma_w = O^n_w x_n \,, \quad \langle \gamma | \gamma \rangle = 1 \,, \tag{88}$$

then we can easily verify that

$$\gamma_w |e^w\rangle = 0 \,, \quad \langle e^w | e^x \rangle = e^2 (\delta_{wx} - \gamma_w \gamma_x) \,. \tag{89}$$

Eq.(84) now takes the form

$$K_{wx} q^2 = \delta_{wx} + M^2 K_{wx} - M^2 K_{wy} \gamma_y \gamma_x \,, \tag{90}$$

from which we readily derive

$$K_{wx} = \frac{1}{q^2} \gamma_w \gamma_x + \frac{1}{q^2 - M^2} (\delta_{wx} - \gamma_w \gamma_x) = \frac{1}{q^2} \gamma_w \gamma_x + \frac{1}{q^2 - M^2} (\rho_w \rho_x + \tau_w \tau_x) \,, \tag{91}$$

where the three unit vectors $\vec{\gamma}$, $\vec{\rho}$ and $\vec{\tau}$ form a complete orthonormal set,[26] with $\gamma_w \rho_x \tau_y \varepsilon^{wxy} = 1$.

Defining

$$A^\mu_\gamma = A^\mu_w \gamma_w \,, \qquad A^\mu_\rho = A^\mu_w \rho_w \,, \qquad A^\mu_\tau = A^\mu_w \tau_w \,, \tag{92}$$

we find immediately that $A^\mu_\gamma$ is strictly conserved since

$$q^\mu A^\mu_\gamma = (q^\mu A^\mu_w) \gamma_w = -i v A_n (e^w)_n \gamma_w = 0 \,. \tag{93}$$

Moreover, again using completeness we can derive, with $|e^r\rangle = r_w |e^w\rangle$:

$$\begin{aligned}
(e^\rho)_n (e^\rho)_k + (e^\tau)_n (e^\tau)_k &= e^2 (\rho_u \rho_y + \tau_u \tau_y + \gamma_u \gamma_y)(e^u)_n (e^y)_k \\
&= e^2 (e^u)_n (e^u)_k = e^2 O^u_a O^u_b \, \varepsilon^{anl} \varepsilon^{bkr} x_\ell x_r = e^2 \varepsilon^{anl} \varepsilon^{akr} x_\ell x_r \\
&= e^2 (\delta_{nk} - x_n x_k) \,.
\end{aligned} \tag{94}$$

---

[25]For $N_t = 2$ all the $f$ matrices would commute.

[26]The vector $\vec{\tau}$ is not to be confused with the tachyonic tadpoles $\tau_n$.

The amplitude can therefore be written as

$$\mathcal{M} = A_h R_h(q) B_h + A_\gamma^\alpha R_\gamma^{\alpha\beta}(q) B_\gamma^\beta + \sum_{\sigma=\rho,\tau} A_\sigma^\alpha R_M^{\alpha\beta}(q) B_\sigma^\beta. \tag{95}$$

with

$$R_\gamma^{\alpha\beta}(q) = -i \frac{g^{\alpha\beta}}{q^2}; \tag{96}$$

and we recognize one Higgs, one massless vector (a 'photon') and two massive vectors. [27]

We now turn to the vertices: Eq.(16) still holds. Because of the form of $\langle e^w | e^x \rangle$ we see that $\gamma$ does not couple to the scalars, while

$$
\begin{array}{c}
\text{[diagram]}
\end{array}
= 
\begin{array}{c}
\text{[diagram]}
\end{array}
= 2i \frac{M^2}{v^2} g^{\alpha\beta}, \qquad
\begin{array}{c}
\text{[diagram]}
\end{array}
= 0. \, [28] \tag{97}
$$

The three-vector coupling can be written as

$$
\begin{array}{c}
\text{[diagram]}
\end{array}
= \gamma_w \rho_x \tau_y h^{wxy} Y(p,\alpha;q,\beta;-p-q,\lambda) = e Y(p,\alpha;q,\beta;-p-q,\lambda). \tag{98}
$$

For the four-vector coupling we have

$$\{aabb\} = \{abab\} = e^2, \quad \{abba\} = -2e^2, \quad (a,b) = (\gamma,\rho),(\gamma,\tau) \text{ or } (\rho,\tau), \tag{99}$$

all other combinations vanishing. Therefore,

$$
\begin{array}{c}
\text{[diagram]}
\end{array}
= -ie^2 \left( 2g^{\alpha\beta} g^{\mu\nu} - g^{\alpha\mu} g^{\beta\nu} - g^{\alpha\nu} g^{\beta\mu} \right). \tag{100}
$$

At this point we can start talking about *charged* vectors, denoted by $W^+$ and $W^-$ (or $+$ and $-$). We define

$$A_\pm^\mu = \frac{1}{\sqrt{2}}(A_\rho^\mu \pm i A_\tau^\mu), \qquad B_\pm^\mu = \frac{1}{\sqrt{2}}(B_\rho^\mu \mp i B_\tau^\mu). \, [29] \tag{101}$$

Then,

$$A_\rho^\alpha R_M^{\alpha\beta}(q) B_\rho^\beta + A_\tau^\alpha R_M^{\alpha\beta}(q) B_\tau^\beta = A_+^\alpha R_M^{\alpha\beta}(q) B_+^\beta + A_-^\alpha R_M^{\alpha\beta}(q) B_-^\beta. \tag{102}$$

Note that which one of the two terms ($W^+$ or $W^-$) actually survives depends on what happens *after* the active vertices. We therefore have to symmetrize over the lines $\rho, \tau$ and also over the $W^+, W^-$ lines. For once replacing wavy lines by smooth lines for readability, and using $\pm$ for $W^\pm$, we therefore write

$$
\begin{array}{c}
\text{[diagram]}
\end{array}
, 
$$
$$
\begin{array}{c}
\text{[diagram]}
\end{array}
\tag{103}
$$

---

[27]In [8] this was employed to arrive at an 'electroweak' model without $Z$ bosons.

[28]We leave the effect of replacing one $h$ by the tadpole, turning the four-point vertex into a three-point one and giving an extra factor $v$, as understood.

[29]The difference in definition for $A_\pm$ and $B_\pm$ is that $A$ emits, and $B$ absorbs the charged vector.

For the $W^+W^-\gamma\gamma$ and $W^+W^-$-Higgs couplings we can follow the same argument. This gives us the Feynman rules

$$
\lambda \,\, {}^{\gamma} \,\, {}^{W^+}_{W^-} = ie\, Y(p,\alpha;q,\beta;-p-q,\lambda),
$$

$$
{}^{W^+}_{\gamma} \,\, {}^{W^-}_{\gamma} = -ie^2 \left( 2g^{\alpha\beta}g^{\mu\nu} - g^{\alpha\mu}g^{\beta\nu} - g^{\alpha\nu}g^{\beta\mu} \right),
$$

$$
{}^{W^+}_{W^-} \,\, {}^{W^+}_{W^-} = ie^2 \left( 2g^{\alpha\beta}g^{\mu\nu} - g^{\alpha\mu}g^{\beta\nu} - g^{\alpha\nu}g^{\beta\mu} \right),
$$

$$
{}^{W^+}_{W^-} \,\, {}^{h}_{h} = 2i\frac{M^2}{v^2}g^{\alpha\beta}. \tag{104}
$$

## 5.2 The Higgs-Kibble model

The HK model arises if we enlarge the tachyon space of the Apollo model to $N_t = 4$. The general form of the $f^w$ matrices is now given by

$$
(f^w)_k^n = e\, O_z^w (g^z)_k^n, \tag{105}
$$

where the orthogonal matrix $O$ is as in the previous section,[30] and, in block notation

$$
g^1 = \begin{pmatrix} S & 0 \\ 0 & S \end{pmatrix}, \quad g^2 = \begin{pmatrix} 0 & \sigma_1 \\ -\sigma_1 & 0 \end{pmatrix}, \quad g^3 = \begin{pmatrix} 0 & \sigma_3 \\ -\sigma_3 & 0 \end{pmatrix}, \tag{106}
$$

where $\sigma_{1,2,3}$ are the Pauli matrices, and $S = i\sigma_2$.[31] The matrices $f$ have the following properties:

$$
[f^w, f^x] = 2e\, \varepsilon^{wxy} f^y, \quad \{f^w, f^x\}_k^n = -2e^2\, \delta^{wx} \delta^{nk}. \tag{107}
$$

Since this implies $\langle e^w | e^x \rangle = e^2 \delta^{wx}$, we find immediately that

$$
K_{wx} = \frac{\delta_{wx}}{q^2 - M^2}. \tag{108}
$$

Next, from

$$
\left( |e^w\rangle \langle e^w| \right) |x\rangle = 0, \quad \left( |e^w\rangle \langle e^w| \right) |e^x\rangle = |e^x\rangle, \tag{109}
$$

it follows that

$$
\delta_{nk} - x_n x_k = \frac{1}{e^2} (e^w)_n (e^w)_k. \tag{110}
$$

We find the amplitude in a straightforward manner:

$$
\mathcal{M} = A_h R_h(q) B_h + \sum_w A_w^\alpha R_M^{\alpha\beta}(q) B_w^\beta, \tag{111}
$$

so that we have three mass-$M$ vectors and one Higgs scalar. Since the three vectors are all on an equal footing, not much would be gained by trying to introduce the notion of 'charge' in the HK model at this point.[32]

---

[30]The three vector bosons are mixed in an arbitrary way.

[31]A derivation is given in appendix C.

[32]This changes once we introduce fermions, see later on.

## 5.3 The electroweak model

We can extend the HK model in the following way. It is possible to choose, in addition to the matrices $f^j$ ($j = 1, 2, 3$) of Eq.(105), a single matrix $f^0$ that commutes with each $f^j$, that is, we can add a single vector boson that has no interactions with the other three.[33] Consistently, we can choose $h^{0wx} = 0$, and the proof of current conservation then goes precisely as before. We also define the vector $\left| e^0 \right\rangle = f^0 \left| x \right\rangle$. This must be a linear combination of the $\left| e^w \right\rangle$, and we write

$$\langle e^w | e^x \rangle = e^2 \delta_{wx}, \quad \left\langle e^0 \middle| e^0 \right\rangle = e'^2, \quad \left\langle e^w \middle| e^0 \right\rangle = ee' z_w, \quad z_w z_w = 1. \tag{112}$$

Application of Eq.(84) then gives, with $M = ev$ and $M' = e'v$,

$$\begin{aligned} K_{wx}(q^2 - M^2) &= \delta_{wx} + MM' K_{w0} z_x, \\ K_{w0}(q^2 - M'^2) &= MM' K_{wx} z_x, \\ K_{00}(q^2 - M'^2) &= 1 + MM' K_{w0} z_w. \end{aligned} \tag{113}$$

After a little algebra[34] we find

$$\begin{aligned} K_{wx} &= \frac{z_w z_x c_\theta^2}{q^2 - N^2} + \frac{z_w z_x s_\theta^2}{q^2} + \frac{\delta_{wx} - z_w z_x}{q^2 - M^2}, \\ K_{w0} &= \frac{z_w c_\theta s_\theta}{q^2 - N^2} - \frac{z_w c_\theta s_\theta}{q^2}, \\ K_{00} &= \frac{s_\theta^2}{q^2 - N^2} + \frac{c_\theta^2}{q^2}, \end{aligned} \tag{114}$$

where we have introduced

$$N^2 = M^2 + M'^2, \quad M = c_\theta N, \, M' = s_\theta N, \quad c_\theta^2 + s_\theta^2 = 1. \tag{115}$$

We are therefore naturally led to define

$$A_Z^\mu = c_\theta A_w^\mu z_w + s_\theta A_0^\mu, \quad A_\gamma^\mu = -s_\theta A_w^\mu z_w + c_\theta A_0^\mu. \tag{116}$$

From Eq.(110) we derive

$$(e^w)_n z_w = \frac{1}{ee'} (e^w)_n \left\langle e^w | e^0 \right\rangle = \frac{e}{e'} (e^0)_n = \frac{c_\theta}{s_\theta} (e^0)_n, \tag{117}$$

and we see that $A_\gamma$ is strictly conserved:

$$q^\mu A_\gamma^\mu \sim -s_\theta A_n (e^w)_n z_w + c_\theta A_n (e^0)_n = 0. \tag{118}$$

We can complement the vector $\vec{z}$ by two unit vectors $\vec{r}$ and $\vec{t}$ into an orthonormal set, so that

$$\delta_{wx} - z_w z_x = r_w r_x + t_w t_x, \tag{119}$$

and define $A_s^\mu = A_w^\mu s_w$ and $\left| e^s \right\rangle = s_w \left| e^w \right\rangle$ ($s = r, t$), so that we can write

$$\delta_{nk} - x_n x_k = \frac{1}{e^2} \left( (e^r)_n (e^r)_k + (e^t)_n (e^t)_k + z_w (e^w)_n z_x (e^x)_k \right). \tag{120}$$

Finally, realizing that

$$A_n (e^w)_n z_w = \frac{i}{v} q^\mu A_w^\mu z_w = \frac{i}{v} q^\mu \left( c_\theta A_Z^\mu - s_\theta A_\gamma^\mu \right) = i \frac{c_\theta}{v} q^\mu A_Z^\mu, \tag{121}$$

---

[33]Only one such matrix $f^0$ can be chosen, see appendix C.

[34]Helped by first working out $K_{wx} z_x$ and then $K_{w0}$.

we find that the amplitude can be written as

$$\mathcal{M} = A_h R_h(q) B_h + \sum_{s=r,t} A_s^\alpha R_M^{\alpha\beta}(q) B_s^\beta + A_Z^\alpha R_N^{\alpha\beta}(q) B_Z^\beta + A_\gamma^\alpha R_\gamma^{\alpha\beta}(q) B_\gamma^\beta \,. \tag{122}$$

We recognize a Higgs scalar, two '$W$' particles of mass $M$, a $Z$ particle of mass $N$, and a massless photon.

Turning to the vertices, we first note that

$$\left| e^\gamma \right\rangle = c_\theta \left| e^0 \right\rangle - s_\theta z_w \left| e^w \right\rangle = 0, \quad \left| e^Z \right\rangle = c_\theta z_w \left| e^w \right\rangle + s_\theta \left| e^0 \right\rangle = \frac{1}{s_\theta} \left| e^0 \right\rangle,$$

$$\left\langle e^Z \middle| e^Z \right\rangle = \frac{N^2}{v^2}, \quad \left\langle e^Z \middle| e^s \right\rangle \sim z_w \left\langle e^w \middle| e^x \right\rangle s_x = 0, \quad \left\langle e^s \middle| e^s \right\rangle = \frac{M^2}{v^2}. \tag{123}$$

Introducing $W^\pm$ as in section 5.1, the nonzero vector-Higgs couplings are

$$\begin{aligned} &\alpha \underset{\beta}{\overset{W^+}{\diagdown}} \overset{h}{\underset{W^-}{\diagup}} = 2ig^{\alpha\beta} \frac{M^2}{v^2}, \qquad \alpha \underset{\beta}{\overset{Z}{\diagdown}} \overset{h}{\underset{Z}{\diagup}} = 2ig^{\alpha\beta} \frac{N^2}{v^2}. \end{aligned} \tag{124}$$

The vector self-interactions are simply obtained by using a factor $c_\theta$ for each $Z$ leg and a factor $-s_\theta$ for each $\gamma$ leg, and remembering that $h^{wxy}$ is now $2e\varepsilon^{wxy}$ rather than $e\varepsilon^{wxy}$: the *weak coupling* is $g = 2e$.

# 6 Inclusion of fermions: General structure

## 6.1 Feynman rules and current conservation

We shall now describe how Dirac fermions can be included in our treatment. We start with massless, chiral fermions, that can be left- or right-handed ($L$ or $R$):

$$\overset{q}{\underset{L}{\longrightarrow}} = \frac{i\,\omega_- \slashed{q}}{q^2}, \qquad \overset{q}{\underset{R}{\longrightarrow}} = \frac{i\,\omega_+ \slashed{q}}{q^2}. \tag{125}$$

Of these fermions (labelled by $a, b, c, \ldots$) there may be any number. The chirality projection operators are

$$\omega_\pm = \tfrac{1}{2}\left(1 \pm \gamma^5\right). \tag{126}$$

The fermions couple to the vectors and to the tachyons with the following Feynman rules:

$$\overset{w}{\underset{}{\wwww}}^\mu = iQ^w \gamma^\mu, \qquad \overset{n}{\underset{}{\diagup}} = -iK^n. \tag{127}$$

The objects $Q^w$ and $K^n$ are matrices in fermion-label space (fl-space), and they have to be Hermitian (see appendix D). Note that $Q^w$ couples $LL$ fermion pairs and $RR$ fermion pairs, while $K^n$ couples to $LR$ and $RL$ pairs. Splitting fl-space into $L$ and $R$ sectors, we therefore have

$$Q^w = \begin{pmatrix} Q_L^w & 0 \\ 0 & Q_R^w \end{pmatrix}, \qquad K^n = \begin{pmatrix} 0 & \kappa_n \\ \kappa_n^\dagger & 0 \end{pmatrix}. \tag{128}$$

Furthermore, it will be handy to distinguish $R$ labels and $L$ labels by dotting the $R$ labels, so that for instance we have $(Q_L^w)_b^a$, $(Q_R^w)_{\dot b}^{\dot a}$, and $(\kappa_n)_{\dot b}^a$. The handlebar rule with fermions is now seen to be [5]

$$\overset{w}{\wwww}\hspace \bigodot = \cdot\overset{w}{\diagup}\bigodot - \cdot\overset{w}{\diagdown}\bigodot, \quad \cdot\overset{w}{\diagup} = \cdot\overset{w}{\diagdown} = iQ^w, \quad \overset{}{\longrightarrow} = i. \tag{129}$$

This rule is only valid for *massless* chiral fermions. Further information can be gleaned from the tree-level four-point amplitudes:

$$\text{[diagrams]} \stackrel{!}{=} 0,$$

$$\text{[diagrams]} \stackrel{!}{=} 0, \tag{130}$$

from which

$$[Q^w, Q^x] \stackrel{!}{=} -ih^{wxy}Q^y, \qquad [Q^w, K^n] \stackrel{!}{=} i(f^w)^n_\ell K^\ell. \tag{131}$$

From the block notation we can write this also as

$$[Q^w_J, Q^x_J] \stackrel{!}{=} -ih^{wxy}Q^y_J \,(J=L,R), \quad Q^w_L\kappa_n - \kappa_n Q^w_R \stackrel{!}{=} i(f^w)^n_\ell \kappa_\ell. \tag{132}$$

The Jacobi identity $[Q^w, [Q^x, K^n]] - [Q^x, [Q^w, K^n]] = [[Q^w, Q^x], K^n]$ provides a consistency check on Eq.(131). While the $K^n$ have to be hermitian, Eq.(131) shows that the $\kappa_n$ cannot all be real.

The equations (130) also allow us to prove current conservation for our models with fermions in an almost trivial manner, by including the fermions appropriately in the SDe:

$$\text{[diagrams]}, \tag{133}$$

## 6.2 Dressed propagators, amplitudes and vertices

There are two passive vertices involving fermions:

$$b \xrightarrow{R} \xrightarrow{L} a = -i(Y)^a_b, \qquad b \xrightarrow{L} \xrightarrow{R} a = -i(Y^\dagger)^{\dot a}_{\dot b}, \quad Y = \tau_n\kappa_n. \tag{134}$$

The matrices $\kappa_n$ are not necessarily square, since we do not have to have equal numbers of $L$ and $R$ fermions.[35] This can be remedied by adding an appropriate number of *barren* fermions,[36] that have no interactions whatsoever (zero entries in the $Q^w$ and $K^n$ matrices) and therefore cannot influence the physics. Having done that we can employ singular value decomposition; that is, we can find unitary matrices $U^a_b$ and $V^{\dot a}_{\dot b}$ in fl-space such that

$$(YY^\dagger)^a_b = (U^\dagger)^a_c D^c_{\dot d}(D^\dagger)^{\dot d}_e U^e_b, \qquad (Y^\dagger Y)^{\dot a}_{\dot b} = (V^\dagger)^{\dot a}_c(D^\dagger)^{\dot c}_d D^d_{\dot e} V^{\dot e}_{\dot b}, \tag{135}$$

---

[35]As in the Weinberg-Salam model without right-handed neutrinos.

[36]Not to be confused with *sterile* ones, that have vanishing $Q$ entries but nonzero $K$ ones.

with

$$D_b^a = m_a \, \delta_{a\dot{b}}, \quad (D^\dagger)_b^{\dot{a}} = m_a \, \delta_{\dot{a}b} \quad \text{(no summation)}, \tag{136}$$

where the $m_a$ are nonnegative and real.[37] We can then write

$$Y_b^a = (U^\dagger)_c^a D_d^c V_b^{\dot{d}}, \qquad D_b^a = U_c^a Y_d^c (V^\dagger)_b^{\dot{d}}. \tag{137}$$

There are four dressed fermion propagators. We can define

$$\Sigma_{LL} = \;\overset{L}{\longrightarrow}\!\!\bigcirc\!\!\overset{}{\longrightarrow}_{L} = \;\overset{}{\longrightarrow}_{L} + \;\overset{L}{\longrightarrow}\!\!\bigcirc\!\!\overset{R \; L}{\longrightarrow}. \tag{138}$$

Again letting the momentum $q$ run from left to right, we find

$$(\Sigma_{LL})_b^a \, q^2 = i\omega_- \slashed{q} \, \delta_{ab} + \omega_- (YY^\dagger)_c^a (\Sigma_{LL})_b^c, \tag{139}$$

so that

$$(U\Sigma_{LL}U^\dagger)_b^a = \frac{i\omega_- \slashed{q}}{q^2 - m_a^2}\delta_{ab}. \tag{140}$$

Similarly, we have

$$\Sigma_{RR} = \;\overset{}{\longrightarrow}\!\!\bigcirc\!\!\overset{}{\longrightarrow}_{R} \;\; \Rightarrow \;\; (V\Sigma_{RR}V^\dagger)_{\dot{b}}^{\dot{a}} = \frac{i\omega_+ \slashed{q}}{q^2 - m_a^2}\delta_{\dot{a}\dot{b}}. \tag{141}$$

For one of the mixing propagators,

$$\Sigma_{LR} = \;\overset{}{\underset{R}{\longrightarrow}}\!\!\bigcirc\!\!\overset{}{\longrightarrow}_{L} = \;\overset{}{\underset{R}{\longrightarrow}}\!\!\bigcirc\!\!\overset{R \; L}{\longrightarrow}, \tag{142}$$

we find

$$(U\Sigma_{LR}V^\dagger)_{\dot{b}}^a = \frac{i\omega_- m_a}{q^2 - m_a^2}\delta_{a\dot{b}}, \tag{143}$$

and for the other mixing propagator we finally have

$$(V\Sigma_{RL}U^\dagger)_b^{\dot{a}} = \frac{i\omega_+ m_a}{q^2 - m_a^2}\delta_{\dot{a}b}. \tag{144}$$

A general amplitude involving a fermion line between active vertices has the form

$$\mathcal{M} = \bigcirc\!\!\overset{L}{\longrightarrow}\!\!\circ\!\!\overset{}{\longrightarrow}_{L}\!\!\bigcirc + \bigcirc\!\!\overset{R}{\longrightarrow}\!\!\circ\!\!\overset{}{\longrightarrow}_{R}\!\!\bigcirc + \bigcirc\!\!\overset{R}{\longrightarrow}\!\!\circ\!\!\overset{}{\longrightarrow}_{L}\!\!\bigcirc + \bigcirc\!\!\overset{L}{\longrightarrow}\!\!\circ\!\!\overset{}{\longrightarrow}_{R}\!\!\bigcirc$$
$$= B_L \Sigma_{LL} A_L + B_R \Sigma_{RR} A_R + B_L \Sigma_{LR} A_R + B_R \Sigma_{RL} A_L. \tag{145}$$

We now introduce unitarily transformed amplitudes:[38]

$$\hat{A}_L = UA_L, \quad \hat{B}_L = B_L U^\dagger, \quad \hat{A}_R = VA_R, \quad \hat{B}_R = B_R V^\dagger. \tag{146}$$

The amplitude then becomes (summing over $a, \dot{a}$)

$$\mathcal{M} = (\hat{B}_L)_a \frac{i\omega_- \slashed{q}}{q^2 - m_a^2}(\hat{A}_L)^a + (\hat{B}_R)_{\dot{a}} \frac{i\omega_+ \slashed{q}}{q^2 - m_a^2}(\hat{A}_R)^{\dot{a}}$$
$$+ (\hat{B}_L)_a \frac{i\omega_- m_a}{q^2 - m_a^2}(\hat{A}_R)^{\dot{a}} + (\hat{B}_R)_{\dot{a}} \frac{i\omega_+ m_a}{q^2 - m_a^2}(\hat{A}_L)^a$$
$$= B_a \frac{i(\slashed{q} + m_a)}{q^2 - m_a^2}A_a, \tag{147}$$

---

[37]Strictly speaking, singular value decomposition also works if the matrix $Y$ is not square. However, in that case $U$ and $V$ have different dimension, and the matrix $D$ is not diagonal. By employing barren fermions we can use the diagonal form (136).

[38]This convention is consistent since $A$ is a spinor, and $B$ is a *conjugate* spinor.

where we have collected the various chiral amplitudes:

$$
\begin{aligned}
A &= \omega_+ \hat{A}_L + \omega_- \hat{A}_R = \omega_+ U A_L + \omega_- V A_R\,, \\
B &= \hat{B}_L \omega_- + \hat{B}_R \omega_+ = B_L U^\dagger \omega_- + B_R V^\dagger \omega_+\,.
\end{aligned}
\tag{148}
$$

We have now combined pairs of massless chiral fermions into massive[39] Dirac fermions, and we have also found how to rewrite active vertices, for instance

$$
\rightarrow i \Big[ \omega_+ (U Q_L^w U^\dagger)^a_b + \omega_- (V Q_R^w V^\dagger)^{\dot a}_{\dot b} \Big] \gamma^\mu\,,
$$

$$
\rightarrow -\frac{i}{v} \Big( \omega_+ (U Y V^\dagger)^a_b + \omega_- (V Y^\dagger U^\dagger)^b_a \Big) = -i \frac{m_a}{v} \delta_{ab}\,,
\tag{149}
$$

the latter result coming from the singular-value decomposition, Eq.(135). This proves that the Higgs is indeed purely scalar, with no pseudoscalar component. The resultant form of the fermion-vector coupling depends, of course, on the model, whereas the fermion-Higgs coupling is universal.

# 7 Inclusion of fermions: Example models

## 7.1 The Abelian Higgs model

In the AH model, we restrict ourselves to $N_t = 2$, in view of the discussion in (3.4). We can dispense with the superscript $w$, and we have $f_k^n = e S_k^n$. Since $Q$ is hermitian, we can diagonalize it, so that we use

$$
(Q_L)^a_b = q_L^a \delta_{ab}\,, \quad (Q_R)^{\dot a}_{\dot b} = q_R^{\dot a} \delta_{\dot a \dot b} \quad \text{(no summation)}\,.
\tag{150}
$$

Eq.(131) can then be cast in the form

$$
[Q,[Q,K^n]] = e^2 K^n \;\Rightarrow\; (q_L^a - q_R^{\dot b})^2 (\kappa_n)^a_{\dot b} = e^2 (\kappa_n)^a_{\dot b} \quad \text{(no summation)}\,.
\tag{151}
$$

We can simply deduce that for all values $a$ and $\dot b$ for which $(\kappa_n)^a_{\dot b}$ does not vanish, all $q_L^a$ must be equal, or all $q_R^{\dot b}$ must be equal, or both;[40] we choose the latter option. Sectors in fl-space that are not connected by nonzero $\kappa_n$ entries are independent.[41] Let us concentrate on one such sector. Here, the $Q_{L,R}$ matrices are proportional to the unit matrix, and we find immediately that

$$
= \frac{i}{2} \Big( (q_L^a + q_R^a) + (q_L^a - q_R^a) \gamma^5 \Big) \gamma^\mu \delta_{ab}\,.
\tag{152}
$$

This form of the AH model is unavoidably parity-violating, since $q_L^a - q_R^a$ cannot vanish. Furthermore, since $(q_a - q_{\dot b})(\kappa_1)^a_{\dot b} = ie(\kappa_2)^a_{\dot b}$, the matrix $Y$ is equal to $v\kappa_1$ up to a complex phase, which is taken care of by absorbing it into $U^\dagger$. The fermion masses are therefore independent of $|x\rangle$, as desired.

A discussion of the next-simplest model, the Apollo model, involves a considerable amount of detail special to that model alone, and we therefore defer it to appendix E.

---

[39]It is of course possible that $m_a = 0$ for some $a$, especially if barren fermions have to be used.

[40]If the $R$ sector, say, contains barren fermions we must have $q_R = 0$. That is indeed the case in the Weinberg-Salam model, where the barren fermions are neutrinos.

[41]Think of the lepton and quark sectors of the Standard Model. The quark sector has no barren fermions.

## 7.2 The Higgs-Kibble model

### 7.2.1 One fermion doublet

We first restrict $n_f$, the number of fermions, to 2, so that $Q_L^w$, $Q_R^w$, and $\kappa_n$ are $2 \times 2$ matrices. Since now we have $h^{wxy} = 2e\varepsilon^{wxy}$, we can choose

$$Q_J^w = -e\,(O_J)_z^w \sigma^z \lambda_J \quad (J = L, R)\,, \tag{153}$$

with $O_J$ an arbitrary but fixed orthogonal matrix as before, and $\lambda_{L,R}$ either zero or one; we then have $(Q_J^w)^2 = e^2 \lambda_J^2$. By applying Eq.(131) twice, we find

$$(Q_L^w)^2 \kappa_n + \kappa_n (Q_R^w)^2 - 2(Q_L^w)\kappa_n(Q_R^w) = e^2 \kappa_n\,, \tag{154}$$

in other words,

$$2(Q_L^w)\kappa_n(Q_R^w) = e^2(\lambda_L^2 + \lambda_R^2 - 1)\kappa_n\,, \tag{155}$$

and applying *this* twice we see that $\lambda_L = \lambda_R = 1$ is not possible. We therefore take $\lambda_L = 1$, $\lambda_R = 0$, so that the right-handed fermions have *no* vector interactions, and for simplicity we take $O_L = 1$. We can again bring $Y$ into diagonal form via $UYV^\dagger = D$. The fermion-vector vertex then has the form

 $$= -ie\,\omega_+ \gamma^\mu \,(U\sigma^w U^\dagger)_b^a\,. \tag{156}$$

We can further streamline the model by transforming the $W$ amplitudes:

$$A_w^\mu \to A_j^\mu = R_j^w A_w^\mu\,, \quad R_j^w = \tfrac{1}{2}\,\mathrm{Tr}\big(U\sigma^w U^\dagger \sigma^j\big) \quad (j = 1, 2, 3)\,. \tag{157}$$

Using the Fierz relations for the $2 \times 2$ Pauli matrices:

$$\mathrm{Tr}(A\sigma^w)\,\mathrm{Tr}(B\,\sigma^w) = 2\,\mathrm{Tr}(AB) - \mathrm{Tr}(A)\,\mathrm{Tr}(B)\,,$$
$$\mathrm{Tr}(A\sigma^w B\,\sigma^w) = 2\,\mathrm{Tr}(A)\,\mathrm{Tr}(B) - \mathrm{Tr}(AB)\,, \tag{158}$$

we can show that $R$ is an orthogonal matrix, whose application does not change the Feynman rules of the vector/scalar sector of the model; and $R_j^w(U\sigma^w U^\dagger) = \sigma^j$. We then have the Feynman rule for the fermion-vector interactions:

$$= -ie\,\omega_+ \gamma^\mu\,(\sigma^j)_b^a\,. \tag{159}$$

The vertex with $j = 3$ is flavour-conserving, and we thus recognise the corresponding vector boson as the neutral one. Furthermore one fermion (the 'up') only emits a $W^+$, and the other one (the 'down') can only emit a $W^-$:

$$= -ie\sqrt{2}\,\omega_+\gamma^\mu \begin{pmatrix} 0 & 1 \\ 0 & 0 \end{pmatrix}_b^a, \qquad = -ie\sqrt{2}\,\omega_+\gamma^\mu \begin{pmatrix} 0 & 0 \\ 1 & 0 \end{pmatrix}_b^a. \tag{160}$$

### 7.2.2 More fermion doublets

We can increase the number of fermions to $n_d$ 'up' and $n_d$ 'down' fermions, so that $n_f = 2n_d$. We choose $Q_L^w = -e\sigma^w \otimes \mathbf{1}$, explicitly:

$$Q_L^1 = \begin{pmatrix} 0 & -e\mathbf{1} \\ -e\mathbf{1} & 0 \end{pmatrix}, \qquad Q_L^2 = \begin{pmatrix} 0 & ie\mathbf{1} \\ -ie\mathbf{1} & 0 \end{pmatrix}, \qquad Q_L^3 = \begin{pmatrix} -e\mathbf{1} & 0 \\ 0 & e\mathbf{1} \end{pmatrix}. \tag{161}$$

The block form refers to the 'up' and 'down' sectors, and $\mathbf{1}$ is the $n_d \times n_d$ unit matrix. Using the representation (106), we then have

$$\kappa_2 = -iQ_L^1\kappa_1, \quad \kappa_3 = -iQ_L^3\kappa_1, \quad \kappa_4 = -iQ_L^2\kappa_1. \tag{162}$$

Therefore,

$$Y = \nu\Sigma(\vec{x})\kappa_1, \quad \Sigma(\vec{x}) = x_1 - ix_2Q_L^1 - ix_3Q_L^3 - ix_4Q_L^2, \tag{163}$$

and $\Sigma$ is unitary. That means that the singular-value decomposition will automatically remove all $x$ dependence in diagonalising $Y$ (because $\Sigma$ will form part of $U$): the fermion masses (and mixings) are independent of $|x\rangle$. There is an important restriction, however: we want the vacuum to be flavour-conserving.[42] Therefore the $Y$ matrix should not mix $u$ and $d$ fermions: it must have a block-diagonal form. We must therefore choose $\kappa_1$ to be (a transformation of) a block-diagonal matrix in fl-space:

$$\nu\kappa_1 = \Sigma(\vec{z})\begin{pmatrix} K_u & 0 \\ 0 & K_d \end{pmatrix}, \tag{164}$$

with $\vec{z}$ an arbitrary unit vector. This kind of restriction of the form of $\kappa_1$ is, in fact, also present in the canonical derivation of the Standard Model, since also there flavour-changing vacuum terms are explicitly excluded, thus forbidding precisely one-half of all possible Yukawa interactions. The matrices $U$ and $V$ are now chosen as

$$U = C\Sigma(\vec{x})^\dagger\Sigma(\vec{z})^\dagger, \quad C = \begin{pmatrix} C_u & 0 \\ 0 & C_d \end{pmatrix}, \quad V = \begin{pmatrix} V_u & 0 \\ 0 & V_d \end{pmatrix}, \tag{165}$$

where

$$C_j K_j V_j^\dagger \quad (j = u, d), \tag{166}$$

is precisely the singular-value decomposition of the $(n_f \times n_f)$ $K$ submatrices. Finally, we define the $R_j^w$ as in Eq.(157), only in the 'tensored' form, using $C$ rather than $U$ and $\sigma^w \otimes \mathbf{1}$. Again considering $W^\pm$ rather than $W^{1,2}$, we arrive at the following vertices:

$$\begin{aligned} &= -ie\sqrt{2}\,\omega_+\gamma^\mu \begin{pmatrix} 0 & C_uC_d^\dagger \\ 0 & 0 \end{pmatrix}_b^a, \\ &= -ie\sqrt{2}\,\omega_+\gamma^\mu \begin{pmatrix} 0 & 0 \\ C_dC_u^\dagger & 0 \end{pmatrix}_b^a, \\ &= -ie\,\omega_+\gamma^\mu \begin{pmatrix} \mathbf{1} & 0 \\ 0 & -\mathbf{1} \end{pmatrix}_b^a. \end{aligned} \tag{167}$$

The matrix $C_uC_d^\dagger$ is, of course, the CKM matrix in the case of quarks, or the PMNS matrix if we consider leptons.[43]

## 7.3 The electroweak model

For simplicity, we shall use $f^w = g^w$ (cf Eq.(106)), and consider only, say, the quark sector of the EW model. Extending the HK model with an additional vector boson, we let the additional vector couple to the fermions, with $Q_L^0$ and $Q_R^0$. Since we have demanded that $h^{0wx} = 0$, $Q_L^0$ must be proportional to the unit matrix. The only way for the commutation relations

$$[Q^0, Q^w] = 0, \ Q_L^w\kappa^n - \kappa^nQ_R^w = i(f^w)_\ell^n\kappa^\ell, \quad w = 0, 1, 2, 3, \tag{168}$$

---

[42]Since there is no photon, there is no notion of an electrically neutral vacuum.

[43]Which implies, of course, the existence of right-handed neutrinos.

to be consistent is to have

$$f^0 = e' \begin{pmatrix} 0 & 0 & 1 & 0 \\ 0 & 0 & 0 & 1 \\ -1 & 0 & 0 & 0 \\ 0 & -1 & 0 & 0 \end{pmatrix}, \tag{169}$$

which provides a stronger constraint on $f^0$ than in the scenario where only vector bosons were included; furthermore, the 'up-down' block form of $Q_{L,R}^0$ must read

$$Q_L^0 = -e' \begin{pmatrix} a_L \otimes \mathbf{1} & 0 \\ 0 & a_L \otimes \mathbf{1} \end{pmatrix}, \qquad Q_R^0 = -e' \begin{pmatrix} a_R \otimes \mathbf{1} & 0 \\ 0 & b_R \otimes \mathbf{1} \end{pmatrix}, \tag{170}$$

with $a_L - a_R = -1$ and $a_L - b_R = +1$.[44] As in the model containing solely vector bosons, we take the combinations $A_\gamma^\mu = c_\theta A_0^\mu - s_\theta A_w^\mu z_w$, $A_Z^\mu = c_\theta A_w^\mu z_w + s_\theta A_0^\mu$, and $A_W^\mu = s_w A_w^\mu$ with $s_w = t_w, r_w$.[45] Since the vector $z_w$ contains the information in which $A^{1,2,3}$ are mixed, we need to perform the same mixing in the $Q_L^w$ matrices. We do so by constructing a rotation matrix $G$, whose rows are the orthonormal vectors $z_w$, $t_w$ and $r_w$, and taking the product $\tilde{R}_x^w = R_y^w G_x^y$. Performing this rotation leaves us with the Feynman rules:

$$f \xrightarrow{\quad\gamma\ \ ^\mu\quad} f = -iQ_f\gamma^\mu, \qquad f \xrightarrow{\quad Z\ \ ^\mu\quad} f = i(v_f + a_f)\gamma^\mu, \tag{171}$$

with

$$\begin{aligned}
Q_u &= s_\theta e\,(a_L + a_R + 1), & Q_d &= s_\theta e\,(a_L + b_R - 1), \\
v_u &= \frac{e}{c_\theta}\left(-c_\theta^2 + s_\theta^2(a_L + a_R)\right), & a_u &= -\frac{e}{c_\theta}, \\
v_d &= \frac{e}{c_\theta}\left(c_\theta^2 + s_\theta^2(a_L + b_R)\right), & a_d &= \frac{e}{c_\theta}.
\end{aligned} \tag{172}$$

Taking into account that the $ffW$ coupling constant $g_w$, defined by $G_F/\sqrt{2} = g_w^2/M^2$, is given by $g_w = e\sqrt{2}$ (from Eq.(167)), it is easily checked that these vertices are, again, precisely those of the standard electroweak model.

# 8 Conclusions

We have shown how to implement the Higgs mechanism in a purely diagrammatic way, working up from the simplest self-interacting tachyon system to the complete electroweak model. In doing so we found that special care has to taken with tadpole-containing diagrams in order to avoid miscounting. We also proved that all theories of the type we studied must contain a Higgs particle [6], and proved the equivalence theorem [7]. The symmetry structure of theories with vector particles arises naturally from the requirement of unitarity, rather than as preordained.

# Acknowledgments

The authors gratefully acknowledge many useful discussions with Oscar Boher Luna and Tom de Wilt, who while master students have worked on several aspects of this research.

---

[44]The lepton sector of the EW is treated the same way, only with different assignments of $a_{L,R}$ and $b_R$.

[45]It is unnecessary to check again strict current conservation for the amplitude $A^0$, since in the proof of section 5.3 the identity of the active vertex does not enter.

# A   Antihermitian matrices

A hermitian matrix has an orthonormal basis of eigenvectors. The following discussion (included here since the result is less well-known) describes the analogous result for *anti*hermitian matrices, that subsume antisymmetric real matrices such as the $f^w$. Let $M$ be an antihermitian matrix; $M^\dagger M$, being by construction hermitian, has an orthonormal basis of eigenvectors. Let $|a_1\rangle$ be such an eigenvector, normalized to unity, with eigenvalue $\lambda$.

If $\lambda = 0$ then

$$\langle a_1 | M^\dagger M | a_1 \rangle = \| M |a_1\rangle \|^2 = 0 , \tag{A.1}$$

so that $M |a_1\rangle = 0$. Any other value of $\lambda$ must be positive, so that we can write $\lambda = z^2$ with $z$ real. For that case we define $|b_1\rangle = (1/z) M |a_1\rangle$. We immediately find that $|b_1\rangle$ is also normalized to unity, and orthogonal to $|a_1\rangle$; furthermore, $M |b_1\rangle = -M^\dagger |b_1\rangle = -z |a_1\rangle$. In the complement of the span of $|a_1\rangle$ and $|b_1\rangle$, $M^\dagger M$ is again hermitian, and we can repeat the process, to find an $|a_2\rangle$ or a pair $|a_2\rangle, |b_2\rangle$, and so on. We find that the vectors $|a_j\rangle, |b_j\rangle$ ($j = 1, 2, \ldots$) are an orthonormal basis, and that $M$ can be written as

$$M = \sum_j z_j \left( |b_j\rangle\langle a_j| - |a_j\rangle\langle b_j| \right) , \tag{A.2}$$

where the sum runs over all nonzero eigenvalue-square-roots $z_j$ and their $|a_j\rangle, |b_j\rangle$ pairs. If the dimension of $M$ is odd, there must be at least one zero eigenvalue.

# B   Current conservation with external vector particles

In the derivation of Eq.(71) we have used the fact that both the $x$ and $y$ lines are axial-gauge propagators, transverse to $n$. That assumption fails if, say, $y$ is an on-shell line, with polarization vector $\epsilon(q)$, for which $q \cdot q = q \cdot \epsilon(q) = 0$ but $n \cdot \epsilon(q)$ does not necessarily vanish.[46] In that case we must write

$$R_n^{\mu\alpha}(p) h^{wxy} \left( \left( p^\alpha p^\beta - p^2 g^{\alpha\beta} \right) - \left( q^\alpha q^\beta - q^2 g^{\alpha\beta} \right) \right) \epsilon^\beta(q) = i h^{wxy} \epsilon^\mu(q) - i h^{wxy} \frac{(n \cdot \epsilon(q))}{(p \cdot n)} p^\mu , \tag{B.1}$$

and the handlebar rule becomes

$$\text{[diagram]} = \text{[diagram]} - i h^{wxy} \frac{(n \cdot \epsilon(q))}{(p \cdot n)} \text{[diagram]} . \tag{B.2}$$

The first diagram on the right fits in with the proof of current conservation for $w$, while the second term is the handlebar for the 'reduced' process, where $w$ and $y$ are stripped away. We can repeat this process until no external vector particles are left. Therefore the proof of current conservation still holds if a finite number of external vector particles is present.

# C   The $g$ matrices of the HK model and the EW model

Let $\sigma_j$ be the Pauli matrices, and let us denote by $S$ the $2 \times 2$ matrix

$$S = i\sigma_2 = \begin{pmatrix} 0 & 1 \\ -1 & 0 \end{pmatrix} . \tag{C.1}$$

---

[46] If both $x$ and $y$ are external, the amplitude vanishes under the handlebar.

The discussion in appendix A shows that we can always write, in block notation,

$$g^1 = e \begin{pmatrix} S & 0 \\ 0 & S \end{pmatrix}, \quad g^2 = \begin{pmatrix} a_2 S & B \\ -B^T & b_2 S \end{pmatrix}, \quad g^3 = \begin{pmatrix} a_3 S & C \\ -C^T & b_3 S \end{pmatrix}, \tag{C.2}$$

with $a_{2,3}$, $b_{2,3}$, $B$ and $C$ to be determined. Let us also write $h^{wxy} = ek\,\varepsilon^{wxy}$, with also $k$ to be determined. The commutator identity of Eq.(75) then implies

$$a_{2,3} = b_{2,3} = 0, \quad [S,B] = kC, \quad [S,C] = -kB. \tag{C.3}$$

Thus we have

$$[S,[S,B]] = -k^2 B \implies \sigma_2 B \sigma_2 = -rB, \quad r = -1 + k^2/2. \tag{C.4}$$

Therefore $B$ must be a linear combination of $\sigma_1$ and $\sigma_3$, and $r = 1$. If we choose $B = e\sigma_1$, then $C = e\sigma_3$, and we arrive at the representation of Eq.(106), with $k = 2$.

For the electroweak model, the matrix $f^0$ that commutes with $g^{1,2,3}$ (and consequently with $f^{1,2,3}$) has the general form

$$f^0 = \begin{pmatrix} e_1 S & e_2 + e_3 S \\ -e_2 + e_3 S & -e_1 S \end{pmatrix}, \quad (f^0)^2 = e_1^2 + e_2^2 + e_3^2 \equiv e'^2 .^{47} \tag{C.5}$$

The numbers $e_{1,2,3}$ can be chosen freely, but $f^0$ matrices with different $e_{1,2,3}$ do not commute with one another. There is therefore room for only *one* extra vector besides the three self-interacting ones in the electroweak model.

## D Hermiticity from cutting rules

Since we do not use Lagrangians or actions, the hermiticity of the $Q$ and $K$ matrices must be argued diagrammatically. To this end we may use the Cutkosky cutting rules [9], that embody the unitarity of a theory. Consider a particular diagram in Quantum Electrodynamics,

$$\tag{D.1}$$

where for now the labels $a, b$ and $c$ are just for telling the fermion lines apart. The Cutkosky rule for this diagram reads

$$\left. \right) + \quad + \quad = 0 ,^{48} \tag{D.2}$$

where the convention is that propagators that are cut by the shaded line are on shell, while all momentum integrations remain. On the left of the shaded line we have the amplitude as it stands, and on the right we have the *complex conjugate* of the *time-reversed* amplitude [5]. Eq.(D.2) can therefore also be written as

$$\left( \right) + \left( \right)\left( \right)^* + \left( \right)^* = 0 . \tag{D.3}$$

For QED, the Cutkosky rule holds for this diagram, but the cancellation is far from trivial [1]. Let us now replace the internal photon line in the diagram (D.1) by the vector particle $w$, and let the electron be replaced by the fermions $a, b, c$ of our model. The terms in Eq.(D.3) then pick up, respectively, the factors

$$(Q^w)_c^a (Q^w)_b^c, \quad (Q^{w\dagger})_c^a (Q^w)_b^c, \quad (Q^{w\dagger})_c^a (Q^{w\dagger})_b^c.$$

---

$^{47}$By explicit calculation of the commutators.

$^{48}$If the quantum numbers of $a$ and $b$ are equal, this is also called the Optical Theorem.

The only reasonable way to still have the null result is to have these three factors equal. By putting $b = a$ and summing, we therefore have

$$\text{Tr}(Q^w Q^w) = \text{Tr}\big(Q^{w\dagger} Q^{w\dagger}\big) = \text{Tr}\big(Q^w Q^{w\dagger}\big) \;\Rightarrow\; \text{Tr}\big((Q^w - Q^{w\dagger})^2\big) = 0, \tag{D.4}$$

which shows that $Q^w$ must be hermitian.[49] Replacing the internal photon by a scalar in diagram (D.1) does also yields a correct cutting rule; this shows that also $K^n$ must be hermitian.

## E  Fermions in the Apollo model

Let us define, in $L, R$-block notation,

$$H = \tau_n K^n = \left(\begin{array}{cc} 0 & Y \\ Y^\dagger & 0 \end{array}\right), \quad R = \left(\begin{array}{cc} U & 0 \\ 0 & V \end{array}\right) \Rightarrow RHR^\dagger = \left(\begin{array}{cc} 0 & D \\ D & 0 \end{array}\right). \tag{E.1}$$

Furthermore, let

$$Q^\gamma = \gamma_w Q^w = \left(\begin{array}{cc} Q_L^\gamma & 0 \\ 0 & Q_R^\gamma \end{array}\right), \qquad \hat{Q}^\gamma = RQ^\gamma R^\dagger = \left(\begin{array}{cc} \hat{Q}_L^\gamma & 0 \\ 0 & \hat{Q}_R^\gamma \end{array}\right). \tag{E.2}$$

Eq.(131) then implies

$$[Q^\gamma, H] = i v \gamma_w x_n (f^w)_\ell^n K^\ell = i v \gamma_w (e^w)_\ell K^\ell = 0. \tag{E.3}$$

Therefore $\hat{Q}^\gamma$ commutes with $RHR^\dagger$, so

$$\hat{Q}_L^\gamma D = D \hat{Q}_R^\gamma, \qquad \hat{Q}_R^\gamma D = D \hat{Q}_L^\gamma. \tag{E.4}$$

This means that both $\hat{Q}_L^\gamma$ and $\hat{Q}_R^\gamma$ commute with $D^2$. If all the fermions masses are different, this means that the $\hat{Q}_{L,R}^\gamma$ are also diagonal in fl-space; if several masses are equal, we can *make* the $\hat{Q}_{L,R}^\gamma$ diagonal by an appropriate orthogonal transformation in fl-space. It is then easily seen that

$$(\hat{Q}_L^\gamma)_a^a = (\hat{Q}_R^\gamma)_a^a, \quad \text{if} \quad m_a \neq 0 \text{ (no summation)}, \tag{E.5}$$

the left- and right-handed couplings are the same for massive fermions. This gives us the fermion-fermion-photon vertex:

$$\begin{aligned} &= i(\hat{Q}_L^\gamma)_a^a \gamma^\mu \delta_{ab}, \quad \text{if} \quad m_a > 0 \text{ (no summation)}, \\ &= i\big[\omega_+(\hat{Q}_L^\gamma)_a^a + \omega_-(\hat{Q}_R^\gamma)_{\dot{a}}^{\dot{a}}\big]\gamma^\mu \delta_{ab}, \quad \text{if} \quad m_a = 0 \text{ (no summation)}. \end{aligned} \tag{E.6}$$

The photon's interaction with massive fermions conserves parity as it should; massless fermions can interact with parity violation without endangering the photon's current conservation.[50] The two other fermion-vector couplings,

$$\hat{Q}^\rho = \rho_w RQ^w R^\dagger, \quad \hat{Q}^\tau = \tau_w RQ^w R^\dagger, \tag{E.7}$$

are not automatically current-conserving under this construction, but since these couple to massive vector bosons that is not required anyway.

---

[49]For an antihermitian matrix $A$, we have $\text{Tr}\big(A^2\big) = -\sum_{a,b} |A_b^a|^2$.

[50]This sidesteps the question of the physical viability of a massless fermion coupling to massless photons: to avoid it the $\kappa_n$ must be chosen with some care.

## E.1 Two fermions

It may be helpful to study a simple example. We can assume two massless fermions of both $L$ and $R$ type, and define

$$(f^w)^n_k = -e\,\epsilon^{wnk}\,, \qquad Q^w_{L,R} = -\frac{e}{2}\sigma^w\,, \qquad \kappa_n = \lambda\,\sigma^n\,. \tag{E.8}$$

In this case, $\gamma_w = x_w$, and we have

$$Q^\gamma_{L,R} = -\frac{e}{2}(x_w\sigma^w)\,, \qquad Y = \lambda\,v\,(x_n\sigma^n)\,. \tag{E.9}$$

We adopt polar coordinates and write $\{\vec{\gamma},\vec{\rho},\vec{\tau}\}$ as

$$\begin{aligned}
\vec{\gamma} &= (\sin(\theta)\cos(\phi)\,,\ \sin(\theta)\sin(\phi)\,,\ \cos(\theta))\,, \\
\vec{\rho} &= (\cos(\theta)\cos(\phi)\,,\ \cos(\theta)\sin(\phi)\,,\ -\sin(\theta))\,, \\
\vec{\tau} &= (-\sin(\phi)\,,\ \cos(\phi)\,,\ 0)\,,
\end{aligned} \tag{E.10}$$

which provides a right-handed orthonormal base,[51] and further we introduce

$$\vec{\sigma}_\pm = \frac{1}{\sqrt{2}}\left(\vec{\rho}\pm i\,\vec{\tau}\right). \tag{E.11}$$

Now, we choose the unitary matrices $U$ and $V$ as follows:

$$\begin{aligned}
U &= \begin{pmatrix} e^{i\phi/2}\cos(\theta/2) & e^{-i\phi/2}\sin(\theta/2) \\ e^{i\phi/2}\sin(\theta/2) & -e^{-i\phi/2}\cos(\theta/2) \end{pmatrix}, \\
V &= \begin{pmatrix} e^{i\phi/2}\cos(\theta/2) & e^{-i\phi/2}\sin(\theta/2) \\ -e^{i\phi/2}\sin(\theta/2) & e^{-i\phi/2}\cos(\theta/2) \end{pmatrix}.
\end{aligned} \tag{E.12}$$

This gives us

$$UYV^\dagger = VYU^\dagger = \lambda\,v\,\mathbf{1}\,, \qquad UQ^\gamma_L U^\dagger = VQ^\gamma_R V^\dagger = \begin{pmatrix} -e/2 & 0 \\ 0 & e/2 \end{pmatrix}. \tag{E.13}$$

We end up with two massive Dirac fermion of mass $\lambda v$ (independent of $|x\rangle$), one with electric charge $-e/2$ and the other with $+e/2$. Turning to charged $W$ bosons, we find

$$(\hat{Q}^+_L)^a_b = (U(\sigma_{+w}Q^w_L)U^\dagger)^a_b = \begin{pmatrix} 0 & e/\sqrt{2} \\ 0 & 0 \end{pmatrix} = -(V(\sigma_{+w}Q^w_R)V^\dagger)^{\dot{a}}_{\dot{b}} = -(\hat{Q}^+_R)^{\dot{a}}_{\dot{b}}\,,$$

$$(\hat{Q}^-_L)^a_b = (U(\sigma_{-w}Q^w_L)U^\dagger)^a_b = \begin{pmatrix} 0 & 0 \\ e/\sqrt{2} & 0 \end{pmatrix} = -(V(\sigma_{-w}Q^w_R)V^\dagger)^{\dot{a}}_{\dot{b}} = -(\hat{Q}^-_R)^{\dot{a}}_{\dot{b}}\,. \tag{E.14}$$

As expected, only one of the fermions can emit a $W^+$, and the other can only emit a $W^-$. To recover the Feynman rules, we consider the emission of a $W^+$ from a fermion line:

$$= i\left(B_a\omega_+\gamma^\mu(\hat{Q}^+_L)^a_b A^b + B_{\dot{a}}\omega_-\gamma^\mu(\hat{Q}^+_R)^{\dot{a}}_{\dot{b}} A^{\dot{b}}\right). \tag{E.15}$$

The couplings of the $W^\pm$ are seen to be purely axial in this model; $A^2$ refers to the emission of a fermion of charge $+e/2$, and $B_1$ to the absorption of a charge $-e/2$ fermion; and of course $A^1$ emits the negative, while $B_2$ absorbs the positive fermion. The Feynman rules are seen to be

$$= i\frac{e}{\sqrt{2}}\gamma^5\gamma^\mu = \qquad . \tag{E.16}$$

---

[51]In the sense that $(x.\sigma)(\rho.\sigma)(\tau.\sigma) = \sigma^1\sigma^2\sigma^3 = i$.

These charge assignments then also automatically lead to the fermion-photon Feynman rules

$$\frac{\gamma^\mu}{{}_{+/2}\quad{}_{+/2}} = i\frac{e}{2}\gamma^\mu, \qquad \frac{\gamma^\mu}{{}_{-/2}\quad{}_{-/2}} = -i\frac{e}{2}\gamma^\mu. \tag{E.17}$$

### E.2 Three fermions

As another example,[52] we choose a model with three massless fermions of both $L$ and $R$ type, and define

$$(f^w)^a_b = -e\,\varepsilon^{wab}, \quad (Q^w_J)^a_b = ie\,\varepsilon^{wab}, \quad (\kappa_n)^a_{\dot{b}} = \lambda\,\varepsilon^{nab}, \tag{E.18}$$

with $J = L, R$ as before; it is easy to verify that these satisfy Eqs.(130) with $h^{wxy} = e\,\varepsilon^{wxy}$. Now $\gamma_w = x_w$, and we again choose the vectors $\vec{g} = \vec{x}, \vec{\rho}$, and $\vec{\tau}$ according to

$$\tau_a = \varepsilon^{abc} x_b \rho_c, \tag{E.19}$$

and define $\sigma_\pm$ as in the previous section. We find

$$Y^a_{\dot{b}} = v\,x_n(\kappa_n)^a_{\dot{b}} = \lambda v(\rho^a\tau_{\dot{b}} - \tau^a\rho_{\dot{b}}), \qquad (Q^\gamma_J)^a_b = x_w(Q^w_J)^a_b = ie(\rho^a\tau_b - \tau^a\rho_b),$$

$$(Q^\rho_J)^a_b = \rho_w(Q^w_J)^a_b = ie(\tau^a x_b - x^a\tau_b), \qquad (Q^\tau_J)^a_b = \tau_w(Q^w_J)^a_b = ie(x^a\rho_b - \rho^a x_b). \tag{E.20}$$

The appropriate choices for $U$ and $V$ are

$$U^a_b = x^a x_b + \rho^a\rho_b - \tau^a\tau_b,$$

$$V^{\dot{a}}_{\dot{b}} = i\nu x^{\dot{a}} x_{\dot{b}} + \rho^{\dot{a}}\tau_{\dot{b}} + \tau^{\dot{a}}\rho_{\dot{b}}, \quad \nu = \pm. \tag{E.21}$$

Note that two alternative forms for $V$ are available. The matrix $D$ now takes the form

$$D^a_{\dot{b}} = (UYV^\dagger)^a_{\dot{b}} = \lambda v(\rho^a\rho_{\dot{b}} + \tau^a\tau_{\dot{b}}) = \lambda v(\sigma^a_+\sigma_{-\dot{b}} + \sigma^a_-\sigma_{+\dot{b}}). \tag{E.22}$$

We have two fermions of mass $\lambda v$, plus one massless fermion. The fermion-photon interactions are

$$(\hat{Q}^\gamma_L)^a_b = (UQ^\gamma_L U\dagger)^a_b = e(\sigma^a_+\sigma_{-b} - \sigma^a_-\sigma_{+b}),$$

$$(\hat{Q}^\gamma_R)^{\dot{a}}_{\dot{b}} = (VQ^\gamma_L V\dagger)^{\dot{a}}_{\dot{b}} = e(\sigma^{\dot{a}}_+\sigma_{-\dot{b}} - \sigma^{\dot{a}}_-\sigma_{+\dot{b}}). \tag{E.23}$$

We are led to define the off-shell amplitudes for a neutral, a positively charged, and a negatively charged fermion as follows:

$$\bigcirc\!\!\!\!\!\overset{0}{\longrightarrow} = (x \cdot A), \quad \bigcirc\!\!\!\!\!\overset{+}{\longrightarrow} = (\sigma_- \cdot A), \quad \bigcirc\!\!\!\!\!\overset{-}{\longrightarrow} = (\sigma_+ \cdot A),$$

$$\overset{0}{\longrightarrow}\!\!\!\!\!\bigcirc = (B \cdot x), \quad \overset{+}{\longrightarrow}\!\!\!\!\!\bigcirc = (B \cdot \sigma_+), \quad \overset{-}{\longrightarrow}\!\!\!\!\!\bigcirc = (B \cdot \sigma_-). \tag{E.24}$$

From

$$\bigcirc\!\!\!\!\!\overset{\gamma^\mu}{\longrightarrow}\!\!\!\!\!\bigcirc = i\left(B_a\omega_+\gamma^\mu(\hat{Q}^\gamma_L)^a_b A^b + B_{\dot{a}}\omega_-\gamma^\mu(\hat{Q}^\gamma_R)^{\dot{a}}_{\dot{b}} A^{\dot{b}}\right)$$

$$= ie\left((B.\sigma_+)\gamma^\mu(\sigma_- A) - (B.\sigma_-)\gamma^\mu(\sigma_+ A)\right), \tag{E.25}$$

we derive the Feynman rules

$$\overset{\gamma^\mu}{{}_+\!\longrightarrow\!{}_+} = ie\gamma^\mu, \qquad \overset{\gamma^\mu}{{}_-\!\longrightarrow\!{}_-} = -ie\gamma^\mu. \tag{E.26}$$

---

[52]In this section, the dotted-undotted index distinction becomes really useful.

The massless fermion is also the neutral one. For the fermion-$W^\pm$ vertices we have to take

$$(\hat{Q}_L^+)_b^a = (U(\sigma_{+w}Q_L^w)U^\dagger)_b^a = -e(x^a\sigma_{-b} - \sigma_-^a x_b),$$
$$(\hat{Q}_R^+)_{\dot{b}}^{\dot{a}} = (V(\sigma_{+w}Q_R^w)V^\dagger)_{\dot{b}}^{\dot{a}} = ve(x^{\dot{a}}\sigma_{-\dot{b}} + \sigma_-^{\dot{a}} x_{\dot{b}}),$$
$$(\hat{Q}_L^-)_b^a = (U(\sigma_{-w}Q_L^w)U^\dagger)_b^a = e(x^a\sigma_{+b} - \sigma_+^a x_b),$$
$$(\hat{Q}_R^-)_{\dot{b}}^{\dot{a}} = (V(\sigma_{-w}Q_R^w)V^\dagger)_{\dot{b}}^{\dot{a}} = ve(x^{\dot{a}}\sigma_{+\dot{b}} + \sigma_+^{\dot{a}} x_{\dot{b}}), \tag{E.27}$$

and in the same way as above we then arrive at the following Feynman rules:[53]

$$\begin{array}{ll} = -ie\Gamma_{-\nu}\gamma^\mu, & = ie\Gamma_\nu\gamma^\mu, \\[2ex] = ie\Gamma_\nu\gamma^\mu, & = -ie\Gamma_{-\nu}\gamma^\mu, \end{array} \tag{E.28}$$

where

$$\Gamma_+ = 1, \quad \Gamma_- = \gamma^5. \tag{E.29}$$

These couplings are purely vector and axial-vector in character, but they depend on the choice of $v$ in Eq.(E.21): two seemingly different models that, however, are based on the same underlying physics. Note that the two alternatives are simply related by simultaneously performing $v \to -v$ and $e \to -e$.

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

---

[53]The fermion charges are counted along the fermion lines; the $W$ charge is counted outgoing.