# Peer review of "The Higgs Mechanism with Diagrams: a didactic approach"

_SciPost Physics Lecture Notes, doi:SciPost Phys. Lect. Notes 88 (2024)_

## Round 1 · Referee Report · Anonymous (Referee 1) · 2024-9-12

Report
Dear Authors,
I am happy to learn that the statement of identity (53) is not the Goldstone equivalence theorem. But of course I never made any claim to that effect. What I did point out is simply that the statement that you describe in footnote 20 of your article as "the fact that longitudinally polarized vector bosons may be replaced by appropriate (combinations of) scalars" and which is indeed "referred to as the Equivalence Theorem" (ET) and needed to prove unitarity is clearly a scale-dependent statement. For instance, in the cited paper of Veltman [6], ET is characterized as the assertion that "for a process which takes place at a COM energy much larger than the mass M of the vector boson the longitudinally polarized vector boson behaves like the Higgs ghost".
With regards to the tachyonic instability issue I also do not see how the more recent remarks made by you in that regard differ from earlier statements. It could be the case however that the above is simply a misunderstanding on my part and that I have not yet fully grasped the more subtle aspects of your approach. For instance, you say that you "do not treat tachyons as particles, but only as a bare propagator, as stated in the beginning of Section 2.1". But in that section a tachyon is defined as an entity which $has$ a "bare scalar propagator with the `wrong' mass term". What kind of entity could that be I wondered? In your response of July 24 it is asserted that, because of the negative mass term, you "are, for now, forced to view the particles partaking in these interactions as tachyons", but that "these vanish from the spectrum" after SSB. But, SSB is a scale-dependent phenomenon and I do not quite see how that would cease to be the case when translated into your (equivalent) formulation.
But perhaps I approach this the wrong way. Perhaps it is necessary to look upon these sorts of semantical issues in a more fluid manner and to hope that they will disappear upon a more thorough understanding of your results.
I am happy to learn that the statement of identity (53) is not the Goldstone equivalence theorem. But of course I never made any claim to that effect. What I did point out is simply that the statement that you describe in footnote 20 of your article as "the fact that longitudinally polarized vector bosons may be replaced by appropriate (combinations of) scalars" and which is indeed "referred to as the Equivalence Theorem" (ET) and needed to prove unitarity is clearly a scale-dependent statement. For instance, in the cited paper of Veltman [6], ET is characterized as the assertion that "for a process which takes place at a COM energy much larger than the mass M of the vector boson the longitudinally polarized vector boson behaves like the Higgs ghost".
With regards to the tachyonic instability issue I also do not see how the more recent remarks made by you in that regard differ from earlier statements. It could be the case however that the above is simply a misunderstanding on my part and that I have not yet fully grasped the more subtle aspects of your approach. For instance, you say that you "do not treat tachyons as particles, but only as a bare propagator, as stated in the beginning of Section 2.1". But in that section a tachyon is defined as an entity which $has$ a "bare scalar propagator with the `wrong' mass term". What kind of entity could that be I wondered? In your response of July 24 it is asserted that, because of the negative mass term, you "are, for now, forced to view the particles partaking in these interactions as tachyons", but that "these vanish from the spectrum" after SSB. But, SSB is a scale-dependent phenomenon and I do not quite see how that would cease to be the case when translated into your (equivalent) formulation.
But perhaps I approach this the wrong way. Perhaps it is necessary to look upon these sorts of semantical issues in a more fluid manner and to hope that they will disappear upon a more thorough understanding of your results.
Recommendation
Publish (meets expectations and criteria for this Journal)

---

## Editorial Decision

published